# Doubly Constrained Fair Clustering

John Dickerson[1,2], Seyed A. Esmaeili[3], Jamie Morgenstern[4], and Claire Jie Zhang[4]

[1]University of Maryland, College Park
[2]Arthur
[3]Simons Laufer Mathematical Sciences Institute
[4]University of Washington

## Abstract

The remarkable attention which fair clustering has received in the last few years has resulted in a significant number of different notions of fairness. Despite the fact that these notions are well-justified, they are often motivated and studied in a disjoint manner where one fairness desideratum is considered exclusively in isolation from the others. This leaves the understanding of the relations between different fairness notions as an important open problem in fair clustering. In this paper, we take the first step in this direction. Specifically, we consider the two most prominent demographic representation fairness notions in clustering: (1) Group Fairness (**GF**), where the different demographic groups are supposed to have close to population-level representation in each cluster and (2) Diversity in Center Selection (**DS**), where the selected centers are supposed to have close to population-level representation of each group. We show that given a constant approximation algorithm for one constraint (**GF** or **DS** only) we can obtain a constant approximation solution that satisfies both constraints simultaneously. Interestingly, we prove that any given solution that satisfies the **GF** constraint can always be post-processed at a bounded degradation to the clustering cost to additionally satisfy the **DS** constraint while the reverse is not true given a solution that satisfies **DS** instead. Furthermore, we show that both **GF** and **DS** are incompatible (having an empty feasibility set in the worst case) with a collection of other distance-based fairness notions. Finally, we carry experiments to validate our theoretical findings.

## 1 Introduction

Algorithms' deployment in consequential settings, from their use in selecting interview candidates during hiring, to criminal justice systems for risk assessment, to making decisions about where public resources should be allocated [39, 28, 11, 18, 12], have led to an explosion in interest of developing classification and regression algorithms designed to have equitable predictions. More recently, these questions have been extended to unsupervised settings, with the recognition that algorithms which behave as subroutines may have downstream impacts on the ability of a system to make equitable decisions. For example, while personalized advertising seems to be a much lower-risk application than those mentioned above, the advertisements in question might pertain to lending, employment, or housing. For this reason, understanding the impact of unsupervised data pre-processing, including dimensionality reduction and clustering, have been of recent interest, despite the fact that many mathematical operationalizations of fairness do not behave nicely under composition [19].

Clustering is a canonical problem in unsupervised learning and arguably the most fundamental. It is also among the classical problems in operations research and used heavily in facility location as well as customer segmentation. As a result, fairness in clustering has also been well-studied

37th Conference on Neural Information Processing Systems (NeurIPS 2023).

in recent years [8]. Much of this work has identified that existing clustering algorithms fail to satisfy various notions of fairness, and introduce special-purpose algorithms whose outcomes do conform to these definitions. As in supervised learning, a list of mathematical constraints have been introduced as notions of fairness for clustering (more than seven different constraints so far). For example, Chierichetti et al. [17] show algorithms that are guaranteed to produce clustering outputs that prevent the under-representation of a demographic group in a given cluster, hence being compliant to the disparate impact doctrine [22]. Brubach et al. [13] show algorithms that bound the probability of assigning two nearby points to different clusters, therefore guaranteeing a measure of community cohesion. Other algorithms for well-motivated fairness notions have also been introduced, such as minimizing the maximum clustering cost per population among the groups [24, 1], assigning distance-to-center values that are equitable [14], and having proportional demographic representation in the chosen cluster centers [32].

These constraints and others may be well-justified in a variety of specific application domains, and *which* are more appropriate will almost certainly depend on the particular application at hand. The dominant approach in the literature has imposed only one constraint in a given setting, though some applications of clustering (which are upstream of many possible tasks) might naturally force us to reconcile these different constraints with one another. Ideally, one would desire a single clustering of the data which satisfies a collection of fairness notions instead of having different clusterings for different fairness notions. A similar question was investigated in fair classification [31, 18] where it was shown that unless the given classification instance satisfies restrictive conditions, the two desired fairness objectives of calibration and balance cannot be simultaneously satisfied. One would expect that such a guarantee would also hold in fair clustering. For various constraints it can be shown that they are in fact at odds with one another. However, it is also worthwhile on the other hand to ask *if some fair clustering constraints are more compatible with one another, and how one can satisfy both simultaneously?*

**Our Contributions:** In this paper, we take a first step towards understanding this question. In particular, we consider two specific group fairness constraints (1) **GF**: The group fair clustering (**GF**) of Chierichetti et al. [17] which roughly states that clusters should have close to population level proportions of each group and (2) **DS**: The diversity in center selection (**DS**) constraint [32] which roughly states that the selected centers in the clustering should similarly include close to population level proportions of each group. We note that although these two definitions are both concerned with group memberships, the fact that they apply at different "levels" (clustered points vs selected centers) makes the algorithms and guarantees that are applicable for one problem not applicable to the other, certainly not in an obvious way. Further, both of these notions are motivated by disparate impact [22] which essentially states that different groups should receive the same treatment. Therefore, it is natural to consider the intersection of both definitions (**GF** + **DS**). We show that by post-processing any solution satisfying one constraint then we can always satisfy the intersection of both constraints. At a more precise level, we show that an $\alpha$-approximation algorithm for one constraint results in an approximation algorithm for the intersection of the constraints with only a constant degradation to approximation ratio $\alpha$. Additionally, we study the degradation in the clustering cost and show that imposing **DS** on a **GF** solution leads to a bounded degradation of the clustering cost while the reverse is not true. Moreover, we show that both **GF** and **DS** are incompatible (having an empty feasible set) with a set of distance-based fairness constraints that were introduced in the literature. Finally, we validate our finding experimentally. Due to the space limits we refer the reader to Appendix C for the proofs as well as further details.

## 2 Related Work

**Fairness in clustering: GF and DS.** In the line of works on group-level fairness, Chierichetti et al. [17] defined *balance* in the case of two groups to require proportion of a group in any cluster to resemble its proportion in input data. They also proposed the method of creating fairlets and then run generic clustering algorithm on fairlets. Bera et al. [9] generalized the notion of balance to multiple groups, and considered when centers are already chosen how to assign points to centers so that each group has *bounded representation* in each cluster. While only assuming probabilistic group assignment, Esmaeili et al. [21] presented algorithms that guarantee cluster outputs satisfy expected group ratio bounds. Also working with bounded representation, Esmaeili et al. [20] showed how to minimize additive violations of fairness constraint while ensuring clustering cost is within given upper

bound. Besides center-based clustering, group fairness constraint is also applied to spectral clustering [33]. Recent work due to Wang et al. [41] speeds up computation in this setting. Ahmadi et al. [2] applied group fairness to correlation clustering. As a recent follow-up, Ahmadian and Negahbani [4] generalized the setting and improved approximation guarantee. Ahmadian et al. [6] and Chhabra and Mohapatra [16] introduced group fairness in the context of hierarchical clustering, with work due to Knittel et al. [35] proposing an algorithm with improved approximation factor in optimizing cost with fairness setting. Diversity in center selection was first studied in Kleindessner et al. [32] for data summarization tasks. The authors presented an algorithms that solves the fair $k$-centers problem with an approximation factor that is exponential in the number of groups and with a running time that is linear in the number of input points. A follow-up work Jones et al. [29] improved the approximation factor to a constant while keeping the run time linear in number of input points. Concerned with both the diversity in selected center and its distortion of the summary, Angelidakis et al. [7] proposed an approximation algorithm for the $k$-centers problem that ensures number of points assigned to each center is lower bounded. Recent work due to Nguyen et al. [37] generalized the problem to requiring group representation in centers to fall in desired range, which is the setting this work is using.

**Other fairness considerations in clustering.** Besides fairness notions already mentioned, other individual fairness notions include requiring individual points to stay close to points that are similar to themselves in output clusters such as that of [34, 13]. The proportionally fair clustering based on points forming coalitions [15]. A notion based on individual fairness that states that points should have centers within a distance $R$ if there are $n/k$ points around it within $R$ [36, 30]. Newer fairness notions on clustering problems were introduced recently in Ahmadi et al. [3] and Gupta and Dukkipati [26]. For the interested reader, we recommend a recent overview due to Awasthi et al. [8] for exhaustive references and an accessible overview of fair clustering research.

## 3   Preliminaries and Symbols

We are given a set of points $\mathcal{C}$ along with a metric distance $d(.,.)$. The set of chosen centers is denoted by $S \subset \mathcal{C}$ and the assignment function (assigning points to centers) is $\phi : \mathcal{C} \to S$. We are concerned with the $k$-center clustering which minimizes the maximum distance between a point and its assigned center. Formally, we have:

$$\min_{S:|S|\leq k,\phi} \max_{j\in\mathcal{C}} d(j, \phi(j)) \tag{1}$$

In the absence of constraints, the assignment function $\phi(.)$ is trivial since the optimal assignment is to have each point assigned to its nearest center. However, when the clustering problem has constraints this is generally not the case.

In fair clustering, each point $j \in \mathcal{C}$ is assigned a color by a function $\chi(j) = h \in \mathcal{H}$ to indicate its demographic group information, where $\mathcal{H}$ is the set of all colors. For simplicity, we assume that each point has one group associated with it and that the total number of colors $m = |\mathcal{H}|$ is a constant. Moreover, the set of points with color $h$ are denoted by $\mathcal{C}^h$. The total number of points is $n = |\mathcal{C}|$ and the total number of points of color $h$ is $n_h = |\mathcal{C}^h|$. It follows that the proportion of color $h$ is $r_h = \frac{n_h}{n}$. Finally, given a clustering $(S, \phi)$, we denote the set of points in the $i^{\text{th}}$ cluster by $C_i$ and the subset of color $h$ by $C_i^h = C_i \cap \mathcal{C}^h$.

We now formally introduce the group fair clustering (**GF**) and the diverse center selection (**DS**) problems:

**Group Fair Clustering [17, 10, 9, 21, 5]:** Minimize objective (1) subject to proportional demographic representation in each cluster. Specifically, $\forall i \in S, \forall h \in \mathcal{H} : \beta_h|C_i| \leq |C_i^h| \leq \alpha_h|C_i|$ where $\beta_h$ and $\alpha_h$ are pre-set upper and lower bounds for the demographic representation of color $h$ in a given cluster.

**Diverse Center Selection [32, 29, 37]:** Minimize objective (1) subject to the set of centers $S$ satisfying demographic representation. Specifically, denoting the number of centers from demographic (color) $h$ by $k_h = |S \cap \mathcal{C}^h|$, then as done in [37] it must satisfy $k_h^l \leq k_h \leq k_h^u$ where $k_h^l$ and $k_h^u$ are lower and upper bounds set for the number of centers of color $h$, respectvily.

Importantly, throughout we have $\forall h \in \mathcal{H} : \beta_h > 0$. Further, for **GF** we consider solutions that could have violations to the constraints as done in the literature [9, 10]. Specifically, a given a solution $(S, \phi)$ has an additive violation of $\rho$ **GF** if $\rho$ is the smallest number such that the following holds: $\forall i \in S, \forall h \in \mathcal{H} : \beta_h |C_i| - \rho \leq |C_i^h| \leq \alpha_h |C_i| + \rho$. We denote the problem of minimizing the $k$-center objective while satisfying both the **GF** and **DS** constraints as **GF+DS**.

**Why Consider GF and DS in Particular?**   There are two reasons to consider the **GF** and **DS** constraints in particular. First, from the point of view of the application both **GF** and **DS** are concerned with demographic (group) fairness. Further, they are both specifically focused on the representation of groups, i.e. the proportions of the groups (colors) in the clusters for **GF** and in the selected center for **DS**. Second, they are both "distance-agnostic", i.e. given a clustering solution one can decide if it satisfies the **GF** or **DS** constraints without having access to the distance between the points.

## 4   Algorithms for GF+DS

### 4.1   Active Centers

We start by observing the fact that if we wanted to satisfy both **GF** and **DS** simultaneously, then we should make sure that all centers are *active* (having non-empty clusters). More precisely, given a solution $(S, \phi)$ then the **DS** constraints should be satisfied further $\forall i \in S : |C_i| > 0$, i.e. every center in $S$ should have some point assigned to it and therefore not forming an empty cluster. The following example clarifies this:

**Example:**   Consider Figure 1. Suppose we have $k = 2$ and we wish to satisfy the **GF** and **DS** constraints with equal red to blue representation. **DS** requires one blue and one red center. Further, each cluster should have $|C_i^{\text{blue}}| = |C_i^{\text{red}}| = \frac{1}{2}|C_i|$ to satisfy **GF**. Consider the following solution $S_1 = \{2, 4\}$ and $\phi_1$ which assigns all points to point 2 including point 4. This satisfies **GF** and **DS**. Since we have one blue center and one red center. Further, the cluster of center 4 has no points and therefore $0 = |C_i^{\text{blue}}| = |C_i^{\text{red}}| = \frac{1}{2}|C_i|$. Another solution would have $S_2 = S_1 = \{2, 4\}$ but with $\phi_2$ assigning points 2 and 3 to center 2 and points 1 and 4 to center 4. This would also clearly satisfy the **GF** and **DS** constraints.

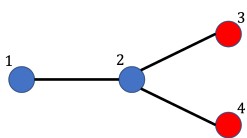

Figure 1: In this graph the distance between the points is defined as the path distance.

There is a clear issue in the first solution which is that although center 4 is included in the selection it has no points assigned to it (it is an empty cluster). This makes it functionally non-existent. This is why the definition should only count active centers.

This issue of active centers did not appear before in **DS** [32, 37], the reason behind this is that it is trivial to satisfy when considering only the **DS** constraint since each center is assigned all the points closest to it. This implies that the center will at least be assigned to itself, therefore all centers in a **DS** solution are *active*. However, we cannot simply assign each point to its closest center when the **GF** constraints are imposed additionally as the colors of the points have to satisfy the upper and lower proportion bounds of **GF**.

### 4.2   The DIVIDE Subroutine

Here we introduce the DIVIDE subroutine (block 1) which is used in subsections 4.3 and 4.4 in algorithms for converting solutions that only satisfy **DS** or **GF** into solutions that satisfy **GF+DS**. DIVIDE takes a set of points $C$ (which is supposed to be a single cluster) with center $i$ along with a subset of chosen points $Q$ ($Q \subset C$). The entire set of points is then divided among the points $Q$ forming $|Q|$ many new non-empty (active) clusters. Importantly, the points of each color are divided among the new centers in $Q$ so that the additive violation increases by at most 2. See Figure 2 for an intuitive illustration.

Here we use the symbol $q$ to index a point in the set $Q$. Importantly, the numbering starts with 0 and ends with $|Q| - 1$.

We prove the following about DIVIDE:

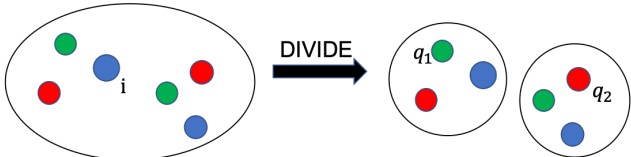

Figure 2: Illustration of DIVIDE subroutine.

---

**Algorithm 1** DIVIDE

---

1: **Input**: Set of points $C$ with center $i \in C$, Subset of points $Q$ ($Q \subset C$) of cardinality $|Q|$.
2: **Output**: An assignment function $\phi : C \to Q$.

3: **if** $|Q| = 1$ **then**
4:     Assign all points $C$ to the single center in $Q$.
5: **else**
6:     Set firstIndex $= 0$.
7:     **for** $h \in \mathcal{H}$ **do**
8:         Set: $T_h = \frac{|C^h|}{|Q|}$, $b_h = T_h - |Q| \lfloor T_h \rfloor$, count $= 0$
9:         Set: $q =$ firstIndex
10:         **while** count $\leq |Q| - 1$ **do**
11:           **if** $b_h > 0$ **then**
12:             Assign $\lceil T_h \rceil$ many points of color $h$ in $C$ to center $q$.
13:             Update $b_h = b_h - 1$.
14:             Update firstIndex $=$ (firstIndex $+ 1$) mod $|Q|$.
15:           **else**
16:             Assign $\lfloor T_h \rfloor$ many points of color $h$ in $C$ to center $q$.
17:           **end if**
18:           Update $q = (q + 1)$ mod $|Q|$, count $=$ count $+ 1$.
19:         **end while**
20:     **end for**
21: **end if**

---

**Lemma 1.** *Given a non-empty cluster $C$ with center $i$ and radius $R$ that satisfies the* **GF** *constraints at an additive violation of $\rho$ and a subset of points $Q$ ($Q \subset C$). Then the clustering $(Q, \phi)$ where $\phi = \text{DIVIDE}(C, Q)$ has the following properties: (1) The* **GF** *constraints are satisfied at an additive violation of at most $\frac{\rho}{|Q|} + 2$. (2) Every center in $Q$ is* active. *(3) The clustering cost is at most $2R$. If $|Q| = 1$ then guarantee (1) is for the additive violation is at most $\rho$.*

### 4.3 Solving GF+DS using a DS Algorithm

Here we show an algorithm that gives a bounded approximation for **GF+DS** using an approximation algorithm for **DS**. Algorithm 2 works by first calling an $\alpha_{\textbf{DS}}$-approximation algorithm resulting in a solution $(\bar{S}, \bar{\phi})$ that satisfies the **DS** constraints, then it solves an assignment problem using the ASSIGNMENTGF algorithm (shown in 3) where points are routed to the centers $\bar{S}$ to satisfy the **GF** constraint. The issue is that some of the centers in $\bar{S}$ may become closed and as a result the solution may no longer satisfy the **DS** constraints. Therefore, we have a final step where more centers are opened using the DIVIDE subroutine to satisfy the **DS** constraints while still satisfying the **GF** constraints at an additive violation and having a bounded increase to the clustering cost.

ASSIGNMENTGF works by solving a linear program (2) to find a clustering which ensures that (1) each cluster has at least a $\beta_h$ fraction and at most an $\alpha_h$ fraction of its points belonging to color $h$, and (2) the clustering assigns each point to a center that is within a minimum possible distance $R$. While the resulting LP solution could be fractional, the last step of ASSIGNMENTGF uses MAXFLOWGF which is an algorithm for rounding an LP solution to valid integral assignments at a bounded degradation to the **GF** guarantees and no increase to the clustering cost. See Appendix B for details on the MAXFLOWGF and its guarantees.

---
**Algorithm 2** DSToGF+DS
---

1: **Input**: Points $\mathcal{C}$, Solution $(\bar{S}, \bar{\phi})$ with clusters $\{C_i, \ldots, C_{\bar{k}}\}$ satisfying the **DS** constraints with $|\bar{S}| = \bar{k} \leq k$ of approximation ratio $\alpha_{\mathbf{DS}}$ for the **DS** clustering problem.
2: **Output**: Solution $(S, \phi)$ satisfying the **GF** and **DS** constraints simultaneously.

3: $(S', \phi') =$ AssignmentGF$(\bar{S}, \mathcal{C})$
4: Update the set of centers $S'$ by deleting all non-active centers (which have no points assigned to them). Let $\{C'_1, \ldots, C'_{k'}\}$ be the (non-empty) clusters of the solution $(S', \phi')$ with $|S'| = k' \leq \bar{k}$.

5: Set $\forall h \in \mathcal{H} : s_h = |S' \cap \mathcal{C}^h|$ , Set $\forall i \in S : Q_i = \{i\}$
6: **while** $\exists h \in \mathcal{H}$ such that $s_h < k_h^l$ **do**
7:     Pick a color $h_0$ such that $s_{h_0} < k_{h_0}^l$.
8:     Pick a center $i \in S'$ where there exists a point of color $h_0$.
9:     Pick a point $j_{h_0}$ of color $h_0$ in cluster $C'_i$
10:    Set $Q_i = Q_i \cup \{j_{h_0}\}$.
11:    Update $s_{h_0} = s_{h_0} + 1$.
12: **end while**
13: **for** $i \in S'$ **do**
14:    $\phi_i =$ Divide$(C'_i, Q_i)$.
15:    $\forall j \in C'_i$ : Set $\phi(j) = \phi_i(j)$.
16: **end for**
17: Set $S = S' \cup \left( \cup_{i \in S'} Q_i \right)$.

---
**Algorithm 3** AssignmentGF
---

1: **Input**: Set of centers $S$, Set of Points $C$.
2: **Output**: An assignment function $\phi : C \rightarrow S$.

3: Using binary search over the distance matrix, find the smallest radius $R$ such that $LP(C, S, R)$ in (2) is feasible and call the solution $\mathbf{x}^*$.
4: Solve MaxFlowGF$(\mathbf{x}^*, C, S)$ and call the solution $\bar{\mathbf{x}}^*$.

---

**LP**$(C, S, R)$ :

$$\forall j \in C, \forall i \in S : x_{ij} = 0 \quad \text{if } d(i, j) > R \tag{2a}$$

$$\forall h \in \mathcal{H}, \forall i \in S : \beta_h \sum_{j \in C} x_{ij} \leq \sum_{j \in C^h} x_{ij} \leq \alpha_h \sum_{j \in C} x_{ij} \tag{2b}$$

$$\forall j \in C : \sum_{i \in S} x_{ij} = 1 \tag{2c}$$

$$\forall j \in C, \forall i \in S : x_{ij} \in [0, 1] \tag{2d}$$

To establish the guarantees we start with the following lemma:

**Lemma 2.** *Solution $(S', \phi')$ of line (3) in algorithm 2 has the following properties: (1) It satisfies the **GF** constraint at an additive violation of 2, (2) It has a clustering cost of at most $(1 + \alpha_{DS})R^*_{GF+DS}$ where $R^*_{GF+DS}$ is the optimal clustering cost (radius) of the optimal solution for **GF+DS**, (3) The set of centers $S'$ is a subset (possibly proper subset) of the set of centers $\bar{S}$, i.e. $S' \subset S$.*

**Theorem 4.1.** *Given an $\alpha_{DS}$-approximation algorithm for the **DS** problem, then we can obtain an $2(1 + \alpha_{DS})$-approximation algorithm that satisfies **GF** at an additive violation of 3 and satisfies **DS** simultaneously.*

**Remark:** If in algorithm 2 no center is deleted in line (4) because it forms an empty cluster, then by Lemma 2 the approximation ratio is $1 + \alpha_{\mathbf{DS}}$ which is an improvement by a factor of 2. Further, the additive violation for **GF** is reduced from 3 to 2.

## 4.4 Solving GF+DS using a GF Solution

---

**Algorithm 4** GFTOGF+DS

---

1: **Input**: Points $\mathcal{C}$, Solution $(\bar{S}, \bar{\phi})$ with clusters $\{\bar{C}_i, \ldots, \bar{C}_{\bar{k}}\}$ satisfying the **GF** constraints with $|\bar{S}| = \bar{k} \le k$.

2: **Output**: Solution $(S, \phi)$ satisfying the **GF** and **DS** constraints simultaneously.

3: Initialize: $\forall h \in \mathcal{H} : s_h = 0, \forall i \in \bar{S} : Q_i = \{\}$.
4: **for** $i \in \bar{S}$ **do**
5:     **if** $\exists h \in \mathcal{H} : s_h < k_h^l$ **then**
6:         Let $h_0$ be a color such that $s_{h_0} < k_{h_0}^l$
7:     **else**
8:         Pick $h_0$ such that $s_{h_0} + 1 \le k_{h_0}^u$.
9:     **end if**
10:     Pick a point $j_{h_0}$ of color $h_0$ in cluster $\bar{C}_i$
11:     Set $Q_i = \{j_{h_0}\}$.
12:     Update $s_{h_0} = s_{h_0} + 1$.
13: **end for**
14: **while** $\exists h \in \mathcal{H} : s_h < k_h^l$ **do**
15:     Pick a color $h_0$ such that $s_{h_0} < k_{h_0}^l$.
16:     Pick a center $i \in \bar{S}$ with cluster $\bar{C}_i$ where there exists a point of color $h_0$ not in $Q_i$.
17:     Pick a point $j_{h_0}$ of color $h_0$ in cluster $\bar{C}_i$
18:     Set $Q_i = Q_i \cup \{j_{h_0}\}$.
19:     Update $s_{h_0} = s_{h_0} + 1$.
20: **end while**
21: Set $S = \cup_{i \in \bar{S}} Q_i$.
22: **for** $i \in \bar{S}$ **do**
23:     $\phi_i = \text{DIVIDE}(\bar{C}_i, Q_i)$.
24:     $\forall j \in \bar{C}_i :$ Set $\phi(j) = \phi_i(j)$. {Assignment to center is updated using DIVIDE.}
25: **end for**

---

Here we start with a solution $(\bar{S}, \bar{\phi})$ of cost $\bar{R}$ that satisfies the **GF** constraints and we want to make it satisfy **GF** and **DS** simultaneously. More specifically, given any **GF** solution we show how it can be post-processed to satisfy **GF+DS** at a bounded increase to its clustering cost by a factor of 2 (see Theorem 4.2). This implies as a corollary that if we have an $\alpha_{\textbf{GF}}$-approximation algorithm for **GF** then we can obtain a $2\alpha_{\textbf{GF}}$-approximation algorithm for **GF+DS** (see Corollary 1).

The algorithm essentially first "covers" each given cluster $\bar{C}_i$ of the given solution $(\bar{S}, \bar{\phi})$ by picking a point of some color $h$ to be a *future* center given that picking a point of such a color would not violate the **DS** constraints (lines(4-13)). If there are still colors which do not have enough picked centers (below the lower bound $k_h^l$), then more points are picked from clusters where points of such colors exist (lines(14-20)). Once the algorithm has picked correct points for each color, then the DIVIDE subroutine is called to divide the cluster among the picked points.

Now we state the main theorem:

**Theorem 4.2.** *If we have a solution $(\bar{S}, \bar{\phi})$ of cost $\bar{R}$ that satisfies the **GF** constraints where the number of non-empty clusters is $|\bar{S}| = \bar{k} \le k$, then we can obtain a solution $(S, \phi)$ that satisfies **GF** at an additive violation of 2 and **DS** simultaneously with cost $R \le 2\bar{R}$.*

**Corollary 1.** *Given an $\alpha_{\textbf{GF}}$-approximation algorithm for **GF**, then we can have a $2\alpha_{\textbf{GF}}$-approximation algorithm that satisfies **GF** at an additive violation of 2 and **DS** simultaneously.*

**Remark:** If the given **GF** solution has the number of cluster $\bar{k} = k$, then the output will have an additive violation of zero, i.e. satisfy the **GF** constraints exactly. This would happen DIVIDE would always receive $Q_i$ with $|Q_i| = 1$ and therefore we can use the guarantee of DIVIDE for the special case of $|Q| = 1$.

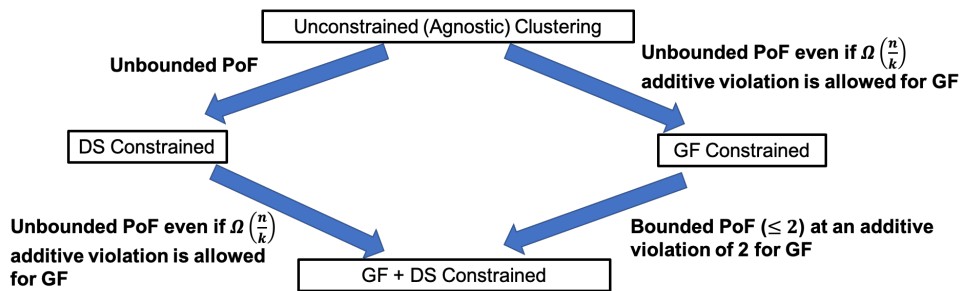

Figure 3: Figure showing the $\mathrm{PoF}$ relation between Unconstrained, **GF**, **DS**, and **GF+DS** clustering.

## 5 Price of (Doubly) Fair Clustering

Here we study the degradation in the clustering cost (the price of fairness) that comes from imposing the fairness constraint on the clustering objective. The price of fairness $\mathrm{PoF}_c$ is defined as $\mathrm{PoF}_c = \frac{\text{Clustering Cost subject to Constraint } c}{\text{Clustering Cost of Agnostic Solution}}$ [20, 9]. Note that since we have two constrains here **GF** and **DS**, we also consider prices of fairness of the form $\mathrm{PoF}_{c_1 \to c_2} = \frac{\text{Clustering Cost subject to Constraints } c_1 \text{ and } c_2}{\text{Clustering Cost subject to Constraint } c_1}$ which equal the amount of degradation in the clustering cost if we were to impose constraint $c_2$ in addition to constraint $c_1$ which is already imposed. Note that we are concerned with the price of fairness in the *worst case*. Interesingly, we find that imposing the **DS** constraint over the **GF** constraint leads to a bounded $\mathrm{PoF}$ if we allow an additive violation of $2$ for **GF** while the reverse is not true even if we allow an additive violation of $\Omega(\frac{n}{k})$ for **GF**.

We find the following:

**Proposition 5.1.** *For any value of $k \geq 2$, imposing **GF** can lead to an unbounded $\mathrm{PoF}$ even if we allow an additive violation of $\Omega(\frac{n}{k})$.*

**Proposition 5.2.** *For any value of $k \geq 3$, imposing **DS** can lead to an unbounded $\mathrm{PoF}$.*

**Proposition 5.3.** *For any value of $k \geq 2$, imposing **GF** on a solution that only satisfies **DS** can lead to an unbounded increase in the clustering cost even if we allow an additive violation of $\Omega(\frac{n}{k})$.*

**Proposition 5.4.** *Imposing **DS** on a solution that only satisfies **GF** leads to a bounded increase in the clustering cost of at most $2$ ($\mathrm{PoF} \leq 2$) if we allow an additive violation of $2$ in the **GF** constraints.*

## 6 Incompatibility with Other Distance-Based Fairness Constraints

In this section, we study the incompatibility between the **DS** and **GF** constraints and a family of distance-based fairness constraints. We note that the results of this section do not take into account the clustering cost and are based only on the feasibility set. That is, we consider more than one constraints simultaneously and see if the feasibility set is empty or not. Two constraints are considered *incompatible* if the intersection of their feasible sets is empty. In some cases we also consider solution that could have violations to the constraints. We present two main findings here and defer the proofs and further details to the Appendix section[1].

**Theorem 6.1.** *For any value $k \geq 2$, the **fairness in your neighborhood** [30], **socially fair** constraint [1, 24] are each incompatible with **GF** even if we allow an additive violation of $\Omega(\frac{n}{k})$ in the **GF** constraint. For any value $k \geq 5$, the **proportionally fair** constraints [15] is incompatible with **GF** even if we allow an additive violation of $\Omega(\frac{n}{k})$ in the **GF** constraint.*

**Theorem 6.2.** *For any value $k \geq 3$, the **fairness in your neighborhood** [30], **socially fair** [1, 24] and **proportionally fair** [15] constraints are each incompatible with **DS**.*

---

[1]Note that socially fair clustering [1, 24] is defined as an optimization problem not a constraint. However, it can be straightforwardly turned into a constraint, see the Appendix for full details.

# 7  Experiments

We use Python 3.9, the `CPLEX` package [38] for solving linear programs and `NetworkX` [27] for max-flow rounding. Further, `Scikit-learn` is used for some standard ML related operations. We use commdity hardware, specifically a MacBook Pro with an Apple M2 chip.

We conduct experiments over datasets from the UCI repository [23] to validate our theoretical findings. Specifically, we use the **Adult** dataset sub-sampled to 20,000 records. Gender is used for group membership while the numeric entries are used to form a point (vector) for each record. We use the Euclidean distance. Further, for the **GF** constraints we set the lower and upper proportion bounds to $\beta_h = (1 - \delta)r_h$ and $\alpha_h = (1 + \delta)r_h$ for each color $h$ where $r_h$ is color $h's$ proportion in the dataset and we set $\delta = 0.2$. For the **DS** constraints, since we do not deal with a large number of centers we set $k_h^l = 0.8 r_h k$ and $k_h^u = r_h k$.

We compare the performance of 5 algorithms. Specifically, we have (1) **COLOR-BLIND:** An implementation of the Gonzalez $k$-center algorithm [25] which achieves a 2-approximation for the unconstrained $k$-center problem. (2) **ALG-GF:** A **GF** algorithm which follows the sketch of [9], however the final rounding step is replaced by an implementation of the MAXFLOWGF rounding subroutine. This algorithm has a 3-approximation for the **GF** constrained instance. (3) **ALG-DS:** An algorithm for the **DS** problem recently introduced by [37] for which also has an approximation of 3. (4) **GFTOGFDS:** An implementation of algorithm 4 where we simply use the **GF** algorithm just mentioned to obtain a **GF** solution. (5) **DSTOGFDS:** Similarly an implementation of algorithm 2 where **DS** algorithm is used as a starting point instead.

Throughout we measure the performance of the algorithms in terms of (1) **PoF:** The price of fairness of the algorithm. Note that we always calculate the price of fairness by dividing by the COLOR-BLIND clustering cost since it solves the unconstrained problem. (2) **GF-Violation:** Which is the maximum additive violation of the solution for the **GF** constraint as mentioned before. (3) **DS-Violation:** Which is simply the maximum value of the under-representation or over-representation across all groups in the selected centers.

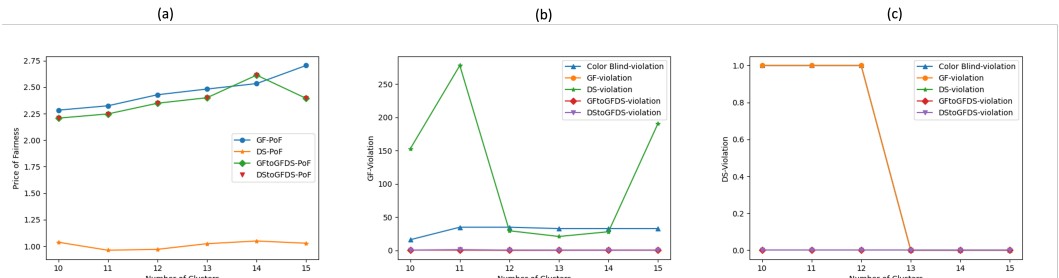

Figure 4: **Adult** dataset results: (a) **PoF** comparison of 5 algorithms, with **COLOR-BLIND** as baseline; (b) **GF-Violation** comparison; (c) **DS-Violation** comparison.

Figure 4 shows the behaviour of all 5 algorithms. In terms of **PoF**, all algorithms have a significant degradation in the clustering cost compared to the COLOR-BLIND baseline except for ALG-DS. However, ALG-DS has a very large **GF-Violation**. In fact, the **GF-Violation** of ALG-DS can be more than 5 times the **GF-Violation** of COLOR-BLIND. This indicates that while ALG-DS has a small clustering cost, it can give very bad guarantees for the **GF** constraints. Finally, in terms of the **DS-Violation** we see that the ALG-GF and the COLOR-BLIND solution can violate the **DS** constraint. Note that both coincide perfectly on each other. Further, although the violation is 1, it is very significant since unlike the **GF** constraints the number of centers can be very small. On the other hand, we see that both GFTOGFDS and DSTOGFDS give the best of both worlds having small values for the **GF-Violation** and zero values for the **DS-Violation** and while their price of fairness can be significant, it is comparable to ALG-GF. Interestingly, the GFTOGFDS and DSTOGFDS are in agreement in terms of measures. This could be because our implementations of the "**GF** part" of DSTOGFDS (its handling of the **GF** constraints) has similarities to the GFTOGFDS algorithm. We show further experiments in the appendix.

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

# A  Useful Facts and Lemmas

**Fact A.1.** *Given non-negative numbers $a_1, a_2, \ldots, a_n$ and positive numbers $b_1, b_2, \ldots, b_n$, then:*

$$\min_{i \in [n]} \frac{a_i}{b_i} \leq \frac{\sum_{i \in [n]} a_i}{\sum_{i \in [n]} b_i} \leq \max_{i \in [n]} \frac{a_i}{b_i}$$

# B  MAXFLOWGF

We start with the following Lemma. First, note that $x_{qj}$ is a decision variable if $x_{qj} = 1$ then point $j$ is assigned to center $q$ and if $x_{qj} = 0$ then it is not. In an integral solutions $x_{qj} \in \{0, 1\}$, but a fractional LP solution could instead have values in $[0, 1]$. We use the bold symbol $\mathbf{x}$ for the collection of value $\{x_{qj}\}_{q \in Q, j \in C}$:

**Lemma 3.** *Given a fractional solution $\mathbf{x}^{frac}$ that satisfies the **GF** constraints at an additive violation of at most $\rho$, then if there exists an integral solution $\mathbf{x}^{integ}$ that satisfies:*

$$\forall q \in Q : \left\lfloor \sum_{j \in C} x_{qj}^{frac} \right\rfloor \leq \sum_{j \in C} x_{qj}^{integ} \leq \left\lceil \sum_{j \in C} x_{qj}^{frac} \right\rceil \tag{3}$$

$$\forall q \in Q, h \in \mathcal{H} : \left\lfloor \sum_{j \in C^h} x_{qj}^{frac} \right\rfloor \leq \sum_{j \in C^h} x_{qj}^{integ} \leq \left\lceil \sum_{j \in C^h} x_{qj}^{frac} \right\rceil \tag{4}$$

*Then this integral solution $\mathbf{x}^{integ}$ satisifies the **GF** constraints at an additive violation of at most $\rho + 2$.*

*Proof.* Since the fractional solution satisfies the **GF** constraints at an additive violation of $\rho$, then we have the following:

$$-\rho + \left( \beta_h \sum_{j \in C} x_{qj}^{\text{frac}} \right) \leq \sum_{j \in C^h} x_{qj}^{\text{frac}} \leq \left( \alpha_h \sum_{j \in C} x_{qj}^{\text{frac}} \right) + \rho$$

We start with the upper bound:

$$\begin{aligned}
\sum_{j \in C^h} x_{qj}^{\text{integ}} &\leq \left\lceil \sum_{j \in C^h} x_{qj}^{\text{frac}} \right\rceil \\
&\leq \sum_{j \in C^h} x_{qj}^{\text{frac}} + 1 \\
&\leq \alpha_h \sum_{j \in C} x_{qj}^{\text{frac}} + \rho + 1 \\
&\leq \alpha_h \Big( \sum_{j \in C} x_{qj}^{\text{integ}} + 1 \Big) + \rho + 1 \\
&\leq \alpha_h \sum_{j \in C} x_{qj}^{\text{integ}} + (\alpha_h + \rho + 1) \\
&\leq \alpha_h \sum_{j \in C} x_{qj}^{\text{integ}} + (\rho + 2)
\end{aligned}$$

Now we do the lower bound:

$$\sum_{j \in C^h} x_{qj}^{\text{integ}} \geq \left\lfloor \sum_{j \in C^h} x_{qj}^{\text{frac}} \right\rfloor$$

$$\geq \sum_{j \in C^h} x_{qj}^{\text{frac}} - 1$$

$$\geq \beta_h \sum_{j \in C} x_{qj}^{\text{frac}} - \rho - 1$$

$$\geq \beta_h (\sum_{j \in C} x_{qj}^{\text{integ}} - 1) - (\rho + 1)$$

$$\geq \beta_h \sum_{j \in C} x_{qj}^{\text{integ}} - (\beta_h + \rho + 1)$$

$$\geq \beta_h \sum_{j \in C} x_{qj}^{\text{integ}} - (\rho + 2)$$

□

The LP solution given to MaxFlowGF satisfies the **GF** constraints at an additive violation of $\rho$, we want to show that the output integral solution satisfies the above conditions of Eqs (3 and 4). The MaxFlowGF($\mathbf{x}^{\text{LP}}, C, Q$) subroutine is similar to that shown in [20, 5, 10]. Specifically, given an LP solution $\mathbf{x}^{\text{LP}} = \{x^{\text{LP}}\}_{q \in Q, j \in Q}$, a set of points $C$, and a set of centers $Q$ and a color assignment function $\chi : C \to \mathcal{H}$ which assigns to each point in $C$ exactly one color in the set of colors $\mathcal{H}$, we construct the flow network $(V, A)$ according to the following:

1. $V = \{s, t\} \cup C \cup \{(q, q^h) | q \in Q, h \in \mathcal{H}\}$.
2. $A = A_1 \cup A_2 \cup A_3 \cup A_4$ where $A_1 = \{(s, j) | j \in C\}$ with upper bound of 1. $A_2 = \{(j, (q, q^h)) | j \in C, x_{qj} > 0\}$ with upper bound of 1. The arc set $A_3 = \{((q, q^h), q) | q \in Q, h \in \mathcal{H}\}$ with lower bound $\left\lfloor \sum_{j \in C^h} x_{qj}^{\text{LP}} \right\rfloor$ and upper bound of $\left\lceil \sum_{j \in C^h} x_{qj}^{\text{LP}} \right\rceil$. As for $A_4 = \{(q, t) | q \in Q\}$ the lower and upper bounds are $\left\lfloor \sum_{j \in C} x_{qj}^{\text{LP}} \right\rfloor$ and $\left\lceil \sum_{j \in C} x_{qj}^{\text{LP}} \right\rceil$.

In the above all lower and upper bounds of the network are integral, therefore if we can show a feasible solution to the above then there must exist an integral flow assignment which also satisfies the constraints. By the construction of the network we have the following fact about any max flow integral solution $\mathbf{x}^{\text{integ}}$.

**Fact B.1.**

$$\forall q \in Q : \left\lfloor \sum_{j \in C} x_{qj}^{LP} \right\rfloor \leq \sum_{j \in C} x_{qj}^{integ} \leq \left\lceil \sum_{j \in C} x_{qj}^{LP} \right\rceil \tag{5}$$

$$\forall q \in Q, h \in \mathcal{H} : \left\lfloor \sum_{j \in C^h} x_{qj}^{LP} \right\rfloor \leq \sum_{j \in C^h} x_{qj}^{integ} \leq \left\lceil \sum_{j \in C^h} x_{qj}^{LP} \right\rceil \tag{6}$$

Accordingly, the following theorem immediately holds:

**Theorem B.1.** *Given an LP solution to* MaxFlowGF *that satisfies the **GF** constraints at an additive violation of $\rho$ and a clustering cost of $R$, then the output integral solution satisfies the **GF** constraints at an additive violation of $\rho + 2$ and a clustering cost of at most $R$.*

*Proof.* The guarantee for the additive violation of **GF** follows immediately from Lemma 3 and Fact B.1. The guarantee for the clustering cost holds, since a point (vertex) $j$ is not connected to a center vertex $(q, q^h)$ unless $x_{qj} > 0$ which can only be the case if $d(j, q) \leq R$. □

## C OMITTED PROOFS

We restate the following lemma and give its proof:

**Lemma 1.** *Given a non-empty cluster $C$ with center $i$ and radius $R$ that satisfies the **GF** constraints at an additive violation of $\rho$ and a subset of points $Q$ ($Q \subset C$). Then the clustering $(Q, \phi)$ where $\phi = \textsc{Divide}(C, Q)$ has the following properties: (1) The **GF** constraints are satisfied at an additive violation of at most $\frac{\rho}{|Q|} + 2$. (2) Every center in $Q$ is active. (3) The clustering cost is at most $2R$. If $|Q| = 1$ then guarantee (1) is for the additive violation is at most $\rho$.*

*Proof.* We first consider the case where $|Q| > 1$. We prove the following claim[2]:

**Claim 1.** *For the fractional assignment $\{x_{qj}^{frac}\}_{q \in Q, j \in C}$ such that:*

$$\forall q \in Q, \forall h \in \mathcal{H} : \sum_{j \in C^h} x_{qj}^{frac} = \frac{|C^h|}{|Q|} = T_h$$

*It holds that: (1) $\forall q \in Q : \sum_{j \in C} x_{qj}^{frac} \geq 1$, (2) **GF** constraints are satisfied at an additive violation of $\frac{\rho}{|Q|}$.*

*Proof.* Now we prove the first property

$$\forall q \in Q : \sum_{j \in C} x_{qj}^{\text{frac}} = \sum_{h \in \mathcal{H}} \sum_{j \in C^h} x_{qj}^{\text{frac}} = \frac{1}{|Q|} \sum_{h \in \mathcal{H}} |C^h| = \frac{|C|}{|Q|} \geq 1 \quad \text{(since } Q \subset C\text{)} \tag{7}$$

.

Since the **GF** constraints given center $i$ are satisfied at an additive violation of $\rho$, then we have:

$$\forall h \in \mathcal{H} : -\rho + \beta_h |C| \leq |C^h| \leq \alpha_h |C| + \rho \tag{8}$$

Therefore, since the amount of color for each center in $Q$ with the fractional assignment can be obtained by dividing by $|Q|$, then we have:

$$\forall h \in \mathcal{H}, \forall q \in Q : -\frac{\rho}{|Q|} + \beta_h \sum_{j \in C} x_{qj}^{\text{frac}} \leq \sum_{j \in C^h} x_{qj}^{\text{frac}} \leq \alpha_h \sum_{j \in C} x_{qj}^{\text{frac}} + \frac{\rho}{|Q|} \tag{9}$$

Therefore the **GF** constraints are satisfied at an additive violation of $\frac{\rho}{|Q|}$. $\qquad\square$

Denoting the assignment $\phi$ resulting from $\textsc{Divide}$ by $\{x_{qj}^{\text{integ}}\}_{q \in Q, j \in C}$, then the following claim holds:

**Claim 2.**

$$\forall q \in Q : \left\lfloor \sum_{j \in C} x_{qj}^{frac} \right\rfloor \leq \sum_{j \in C} x_{qj}^{integ} \leq \left\lceil \sum_{j \in C} x_{qj}^{frac} \right\rceil$$

$$\forall q \in Q, h \in \mathcal{H} : \left\lfloor \sum_{j \in C^h} x_{qj}^{frac} \right\rfloor \leq \sum_{j \in C^h} x_{qj}^{integ} \leq \left\lceil \sum_{j \in C^h} x_{qj}^{frac} \right\rceil$$

*Proof.* For any color $h$ we have $|C_h| = a_h |Q| + b_h$ where $a_h$ and $b_h$ are non-negative integers and $b_h$ is the remainder of dividing $|C_h|$ by $Q$ ($b_h \in \{0, 1, \ldots, |Q| - 1\}$). It follows that $\sum_{j \in C^h} x_{qj}^{\text{frac}} = T_h = a_h + \frac{b_h}{|Q|}$. $\textsc{Divide}$ gives each center either $\sum_{j \in C^h} x_{qj}^{\text{integ}} = a_h = \lfloor T_h \rfloor = \left\lfloor \sum_{j \in C^h} x_{qj}^{\text{frac}} \right\rfloor$ or $\sum_{j \in C^h} x_{qj}^{\text{integ}} = a_h + 1 = \lceil T_h \rceil = \left\lceil \sum_{j \in C^h} x_{qj}^{\text{frac}} \right\rceil$. This proves the second condition.

For the first condition, note that $|C| = \sum_{h \in \mathcal{H}} (a_h |Q| + b_h) = (\sum_{h \in \mathcal{H}} a_h)|Q| + a|Q| + b$ where we set $\sum_{h \in \mathcal{H}} b_h = a|Q| + b$ with $a$ and $b$ being non-negative integers. $b$ is the remainder and has values

---

[2]In our notation $x_{qj} \in [0, 1]$ denotes the assignment of point $j$ to center $q$.

in $\{0, 1, \ldots, |Q| - 1\}$. Accordingly, the sum of the remainders across the colors is $a|Q| + b$. Since the remainders are added "successivly" across the centers (see Figure 2) and $a$ is divisible by $|Q|$, then for any center $q \in Q$ either $\sum_{j \in C} x_{qj}^{\text{integ}} = (\sum_{h \in \mathcal{H}} a_h) + a$ or $\sum_{j \in C} x_{qj}^{\text{integ}} = (\sum_{h \in \mathcal{H}} a_h) + a + 1$. Note that $\sum_{j \in C} x_{qj}^{\text{frac}} = \sum_{h \in \mathcal{H}} T_h = (\sum_{h \in \mathcal{H}} a_h) + a + \frac{b}{|Q|}$. Therefore, $\left\lfloor \sum_{j \in C} x_{qj}^{\text{frac}} \right\rfloor = (\sum_{h \in \mathcal{H}} a_h) + a$ and $\left\lceil \sum_{j \in C} x_{qj}^{\text{frac}} \right\rceil = (\sum_{h \in \mathcal{H}} a_h) + a + 1$. This proves, the first condition. $\qquad \square$

By Claim 2 and Lemma 3 it follows that for each center $q \in Q$ the assignment $\{x_{qj}^{\text{integ}}\}_{q \in Q, j \in C}$ satisfies the **GF** constraints at an additive violation of $\frac{\rho}{|Q|} + 2$, this proves the first guarantee.

By Claim 2 and guarentee (1) of Claim 1, then $\forall q \in Q : \sum_{j \in C} x_{qj}^{\text{integ}} \geq \left\lfloor \sum_{j \in C} x_{qj}^{\text{frac}} \right\rfloor \geq 1$. Therfeore, every center $q \in Q$ is *active* proving the second guarantee.

Guarantee (3) follows since $\forall j \in C : d(j, \phi(j)) \leq d(j, i) + d(i, \phi(j)) \leq 2R$.

Now if $|Q| = 1$, then guarantee (2) follows since the cluster $C$ is non-empty. Guarantee (3) follows similarly to the above. The additive violation in the **GF** constraint on the other hand is $\rho$ since the single center $Q$ has the exact set of points that were assigned to the original center $i$. $\qquad \square$

We restate the next lemma and give its proof:

**Lemma 2.** *Solution $(S', \phi')$ of line (3) in algorithm 2 has the following properties: (1) It satisfies the **GF** constraint at an additive violation of 2, (2) It has a clustering cost of at most $(1 + \alpha_{DS}) R_{GF+DS}^*$ where $R_{GF+DS}^*$ is the optimal clustering cost (radius) of the optimal solution for **GF+DS**, (3) The set of centers $S'$ is a subset (possibly proper subset) of the set of centers $\bar{S}$, i.e. $S' \subset S$.*

*Proof.* We begin with the following claim which shows that there exists a solution that only uses centers from $\bar{S}$ to satisfy the **GF** constraints exactly and at a radius of at most $(1 + \alpha_{DS}) R_{GF+DS}^*$. Note that this claim has non-constructive proof, i.e. it only proves the existence of such a solution:

**Claim 3.** *Given the set of centers $\bar{S}$ resulting from the $\alpha_{DS}$-approximation algorithm, then there exists an assignment $\phi_0$ from points in $\mathcal{C}$ to centers in $\bar{S}$ such that the following holds: (1) The **GF** constraint is exactly satisfied (additive violation of 0). (2) The clustering cost is at most $(1 + \alpha_{DS}) R_{GF+DS}^*$.*

*Proof.* Let $(S_{GF+DS}^*, \phi_{GF+DS}^*)$ be an optimal solution to the **GF+DS** problem. $\forall i \in S_{GF+DS}^*$ let $N(i) = \arg \min_{\bar{i} \in \bar{S}} d(i, \bar{i})$, i.e. $N(i)$ is the nearest center in $\bar{S}$ to center $i$ (ties are broken using the smallest index). $\phi_0$ is formed by assigning all points which belong to center $i \in S_{GF+DS}^*$ to $N(i)$. More formally, $\forall j \in \mathcal{C} : \phi_{GF+DS}^*(j) = i$ we set $\phi_0(j) = N(i)$. Note that it is possible for more than one center $i$ in $S_{GF+DS}^*$ to have the same nearest center in $\bar{S}$. We will now show that $\phi_0$ satisfies the **GF** constraint exactly. Note first that if a center $\bar{i} \in \bar{S}$ has not been assigned any points by $\phi_0$, then it is empty and trivially satisfies the **GF** constraint exactly. Therefore, we assume that $\bar{i}$ has a non-empty cluster. Denote by $N^{-1}(\bar{i})$ the set of centers $i \in S_{GF+DS}^*$ for which $\bar{i}$ is the nearest center, then using Fact A.1 and the fact that every cluster in $(S_{GF+DS}^*, \phi_{GF+DS}^*)$ satisfies the **GF** constraint exactly we have:

$$\beta_h \leq \min_{i \in N^{-1}(\bar{i})} \frac{|C_i^h|}{|C_i|} \leq \frac{\sum_{i \in N^{-1}(\bar{i})} |C_i^h|}{\sum_{i \in N^{-1}(\bar{i})} |C_i|} = \frac{|C_{\bar{i}}^h|}{|C_{\bar{i}}|} \leq \max_{i \in N^{-1}(\bar{i})} \frac{|C_i^h|}{|C_i|} \leq \alpha_h \qquad (10)$$

The proves guarantee (1) of the lemma. Now we prove guarantee (2), we denote by $R_{DS}^*$ the optimal clustering cost for the **DS** constrained problem. We can show that $\forall j \in \mathcal{C}$:

$$
\begin{aligned}
d(j, \phi_0(j)) &\leq d(j, \phi_{GF+DS}^*(j)) + d(\phi_{GF+DS}^*(j), \phi_0(j)) \\
&\leq d(j, \phi_{GF+DS}^*(j)) + d(\phi_{GF+DS}^*(j), N(\phi_{GF+DS}^*(j))) \quad (\text{since } \phi_0(j) = N(\phi_{GF+DS}^*(j))) \\
&\leq R_{GF+DS}^* + \alpha_{DS} R_{DS}^* \qquad\qquad\qquad\qquad\qquad (\text{since } \bar{S} \text{ is an } \alpha_{DS}\text{-approximation for } \textbf{DS}) \\
&\leq (1 + \alpha_{DS}) R_{GF+DS}^*
\end{aligned}
$$

Where the last holds since $R_{DS}^* \leq R_{GF+DS}^*$ because the set of solutions constrained by **DS** is a subset of the set of solutions constrained by **GF+DS**. $\qquad \square$

Now we can prove the lemma. By the above claim, it follows that when AssignmentGF is called, the LP solution from line (3) of algorithm block 3 satisfies: (1) The **GF** constraints exactly and (2) Has a clustering cost of at most $(1 + \alpha_{\textbf{DS}})R^*_{\textbf{GF+DS}}$. This is because LP (2) includes all integral assignments from $\mathcal{C}$ to $\bar{S}$ including $\phi_0$. Since this LP assignment is fed to MaxFlowGF it follows by Theorem B.1 that the final solution satisfies: (1) The **GF** constraint at an additive violation of 2, (2) Has a clustering cost of at most $(1 + \alpha_{\textbf{DS}})R^*_{\textbf{GF+DS}}$. Guarantee (3) holds since some centers may become closed (assigned no points) and therefore $S' \subset \bar{S}$ (possibly being a proper subset). $\qquad\square$

We restate the following theorem and give its proof:

**Theorem 4.1.** *Given an $\alpha_{\textbf{DS}}$-approximation algorithm for the **DS** problem, then we can obtain an $2(1 + \alpha_{\textbf{DS}})$-approximation algorithm that satisfies **GF** at an additive violation of 3 and satisfies **DS** simultaneously.*

*Proof.* By Lemma 2 above, the set of centers $S'$ is a subset (possibly proper) subset of $S$ and therefore the **DS** constraints may no longer be satisfied. Algorithm 2 select points from each color $h$ so that when they are added to $S'$, then for each color $h$ the set of centers is at least $\beta_h k$. Since these new centers are opened using the Divide subroutine then it follows that they are all active (guarantee (2) of Lemma 1).

Further, by guarantee (3) of Lemma 1 for Divide we have for any point $j$ assigned to a new center $q$ that $d(j, q) \le 2d(j, \phi'(j)) \le 2(1 + \alpha_{\textbf{DS}})R^*_{\textbf{GF+DS}}$.

Finally, by guarantee (1) of Lemma 1 Divide is called over a cluster that satisfies **GF** at an additive violation of 2 and therefore the resulting additive violation is at most $\max\{2, \frac{2}{|Q_i|} + 2\}$. Since $2 \le \frac{2}{|Q_i|} + 2 \le \frac{2}{2} + 2 = 3$. The additive violation is at most 3. $\qquad\square$

We restate the next theorem and give its proof:

**Theorem 4.2.** *If we have a solution $(\bar{S}, \bar{\phi})$ of cost $\bar{R}$ that satisfies the **GF** constraints where the number of non-empty clusters is $|\bar{S}| = \bar{k} \le k$, then we can obtain a solution $(S, \phi)$ that satisfies **GF** at an additive violation of 2 and **DS** simultaneously with cost $R \le 2\bar{R}$.*

*Proof.* We point out the following fact:

**Fact C.1.** *Every cluster in $(\bar{S}, \bar{\phi})$ has at least one point from each color.*

*Proof.* This holds, since given a center $i \in \bar{S}$ we have $|\bar{C}_i| > 0$ and therefore $\forall h \in \mathcal{H} : |\bar{C}_i^h| \ge \beta_h|\bar{C}_i| > 0$ and therefore $|\bar{C}_i^h| \ge 1$ since it must be an integer. $\qquad\square$

We note that the values $\{\beta_h, \alpha_h\}_{h \in \mathcal{H}}$ and $k$ must lead to a feasible **DS** problem, i.e. there exist positive integers $g_h$ such that $\sum_{h \in \mathcal{H}} g_h = k$ and $\forall h \in \mathcal{H} : \beta_h k \le g_h \le \alpha_h k$. Accordingly, since lines (4-13) in algorithm 4 can always pick a point of some color $h$ such that the upper bound $\alpha_h k$ is not exceeded for every cluster $i$. Therefore the following fact must hold

**Fact C.2.** *By the end of line (13) we have $\forall i \in \bar{S} : |Q_i| \ge 1$.*

Further, the final $s_h$ values are valid for **DS**:

**Claim 4.** *By the end of line (13) the values of $s_h$ satisfy: (1) $\sum_{h \in \mathcal{H}} s_h \le k$, (2) $\forall h \in \mathcal{H} : \beta_h k \le s_h \le \alpha_h k$.*

*Proof.* Lines (4-13) add values to $s_h$ if the lower bound $\beta_h k$ for color $h$ is not satisfied. If the lower bound is satisfied for all colors, then points of some color $h$ are added provided that adding them would not exceed the upper bound of $\alpha_h k$ (see line 5). Therefore, by the end of line (13) for any color $h \in \mathcal{H} : s_h \le \alpha_h k$ and either $s_h \ge \beta_h k$ or $s_h < \beta_h k$[3].

If by the end of line (13) we have $\forall h \in \mathcal{H} : s_h \ge \beta_h k$, then the algorithm moves to line (22). Otherwise, it will keep picking points and incrementing $s_h$ until $\forall h \in \mathcal{H} : s_h \ge \beta_h k$.

---

[3]To see why we could have $s_h < \beta_h k$, consider the case where $\bar{k} < k$ and therefore there would not be enough clusters to so that we can add points for each color.

Further, since such valid **DS** values exist it must be that the above satisfies $\sum_{h \in \mathcal{H}} s_h \le k$ and $\forall h \in \mathcal{H} : s_h \le \alpha_h k$. This concludes the proof for the claim. $\qquad\square$

By Lemma 1 for DIVIDE the new centers $S = \cup_{i \in \bar{S}} Q_i$ are all active (guarantee 2 of DIVIDE) and since the values of $s_h$ are valid (Claim 4 above), therefore $S$ satisfies the **DS** constraints.

Since the assignment in each cluster in the new solution $(S, \phi)$ is formed using DIVIDE over the clusters of $(\bar{S}, \bar{\phi})$ then by guarantee 1 of DIVIDE, each cluster $(S, \phi)$ satisfies **GF** at an additive violation of 2. Finally, the clustering cost is at most $R \le 2\bar{R}$ (guarantee 3 of DIVIDE). $\qquad\square$

We restate the following corollary and give its proof:

**Corollary 1.** *Given an $\alpha_{\textbf{GF}}$-approximation algorithm for **GF**, then we can have a $2\alpha_{\textbf{GF}}$-approximation algorithm that satisfies **GF** at an additive violation of $2$ and **DS** simultaneously.*

*Proof.* Using the previous theorem (Theorem 4.2) the solution $(\bar{S}, \bar{\phi})$ has a cost of $\bar{R} \le \alpha_{\textbf{GF}} \text{OPT}_{\textbf{GF}}$. The post-processed solution that satisfies **GF** at an additive violation of 2 and **DS** simultaneously has a cost of $R \le 2\bar{R} \le 2\alpha_{\textbf{GF}} \text{OPT}_{\textbf{GF}} \le 2\alpha_{\textbf{GF}} \text{OPT}_{\textbf{GF+DS}}$. The last inequality follows because $\text{OPT}_{\textbf{GF}} \le \text{OPT}_{\textbf{GF+DS}}$ which is the case since both problems minimize the same objective, however by definition the constraint set of **GF** + **DS** is a subset of the constraint set of **GF**. $\qquad\square$

Before we proceed, we define the following clustering instance which will be used in the proof:

**Definition 1.** *$\ell$-Community Instance: The $\ell$-community instance is a clustering instance where the set of points $\mathcal{C}$ can be partitioned into $\ell$ communities (subsets) $\{C_1^{CI}, \ldots, C_\ell^{CI}\}$ of coinciding points (points within the same community are separated by a distance of $0$). Further, the communities are of equal size, i.e. $\forall i \in \ell : |C_i^{CI}| = \frac{n}{\ell}$ . Moreover, the distance between any two points belonging to different communities in the partition is at least $R > 0$.*

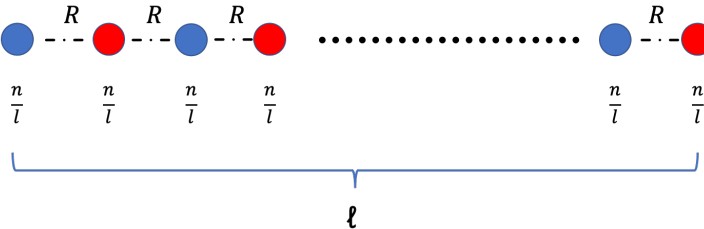

Figure 5: An $\ell$-community instance to show Price of Fairness (**GF**) and incompatibility between **GF** and other fairness constraints when $k$ is even.

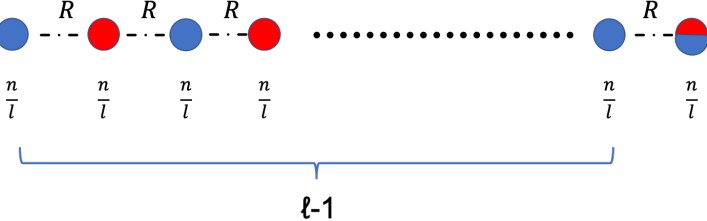

Figure 6: An $\ell$-community instance to show Price of Fairness (**GF**) and incompatibility between **GF** and other fairness constraints when $k$ is odd.

Figures 5 and 6 show two examples of the $\ell$-community instances. When clustering with a value of $k$, the given $\ell$-community instance with $k = \ell$ is arguably the most "natural" clustering instance where the clustering output is the communities $\{C_1^{CI}, \ldots, C_k^{CI}\}$.

The following fact clearly holds for any $\ell$-community instance:

**Fact C.3.** *If we cluster an $\ell$-community instance with $k = \ell$ then: (1) The set of optimal solutions are $(S_{CI}, \phi_{CI})$ where $S_{CI}$ has exactly one center from each community $\{C_1^{CI}, \ldots, C_k^{CI}\}$. Further, points are assigned to a center in the same community. (2) Clustering cost of $(S_{CI}, \phi_{CI})$ is 0. (3) Any solution other than $(S_{CI}, \phi_{CI})$ has a clustering cost of at least $R > 0$.*

We restate the following proposition and give its proof:

**Proposition 5.1.** *For any value of $k \geq 2$, imposing **GF** can lead to an unbounded PoF even if we allow an additive violation of $\Omega(\frac{n}{k})$.*

*Proof.* Consider the case where $k \geq 2$ is even and refer to Figure 5 where we have $\ell = k$ communities that alternate from red to blue color. Further, by Fact C.3 the optimal solution has a clustering cost of 0. The optimal solution would have one center in each of the $k = \ell$ communities, and assign points to its closest center.

If we set the lower and upper proportion bounds to $\frac{1}{2}$ for both colors, then to satisfy **GF** each cluster should have both red and blue points. There must exists a cluster $C_i$ of size $|C_i| \geq \frac{n}{k}$, it follows that to satisfy the **GF** constraints at an additive violation of $\rho$, then $|C_i^{\text{blue}}| \geq \frac{1}{2}|C_i| - \rho = \frac{n}{2k} - \rho$ and similarly we would have $|C_i^{\text{red}}| \geq \frac{n}{2k} - \rho$. By setting $\rho = \frac{n}{2k} - \epsilon$ for some constant $\epsilon > 0$, then we have $|C_i^{\text{blue}}|, |C_i^{\text{red}}| > 0$. This implies that a point will be assigned to a center at a distance $R > 0$ and therefore the PoF is unbounded.

For a value of $k$ that is odd, see the example of Figure 6. Here instead the last community has the same number of red and blue points. We call the cluster whose center is in the last community $C_{\text{last}}$. If $|C_{\text{last}}| \neq \frac{n}{k}$, then there are points assigned to the center of $C_{\text{last}}$ from other communities incurring cost $R > 0$ or points in the last community are assigned to other centers at distance $R > 0$. If $|C_{\text{last}}| = \frac{n}{k}$, then in the remaining $k - 1$ communities with total of $n - \frac{n}{k}$ points, $k - 1$ centers are chosen. There must exists a cluster $C_i$ of size $|C_i| \geq \frac{n}{k}$. We then follow the same argument as in the even $k$ case, which is to satisfy **GF** with additive violation $\rho = \frac{n}{2k} - \epsilon$ for both color, we must have $|C_i^{\text{blue}}|, |C_i^{\text{red}}| > 0$. This means at least a point will be assigned to a center at a distance $R > 0$ and therefore the PoF is unbounded for the odd $k$ case as well. $\qquad\square$

We restate the following proposition and give its proof:

**Proposition 5.2.** *For any value of $k \geq 3$, imposing **DS** can lead to an unbounded PoF.*

*Proof.* Consider a case of the $\ell$ community instance shown in Figure 7 where $k \geq 3$ and $k = \ell$. Here all communities are blue, except for the last which has $\frac{n}{2k}$ red points and $\frac{n}{2k}$ green points. Similar to the previous proposition since it is a community instance with $\ell = k$, by Fact C.3 the optimal solution has a clustering cost of 0 and would have one center in each community and assign each point to its closest center.

Suppose for **DS** we set $k_{\text{blue}}^l, k_{\text{red}}^l, k_{\text{green}}^l > 0$, this implies that we should pick a center of each color. This implies that we can have at most $k - 2$ blue center, therefore there will be a community (composed of all blue points) where no point is picked as a center. Therefore, the clustering cost is $R > 0$ and the PoF is unbounded. $\qquad\square$

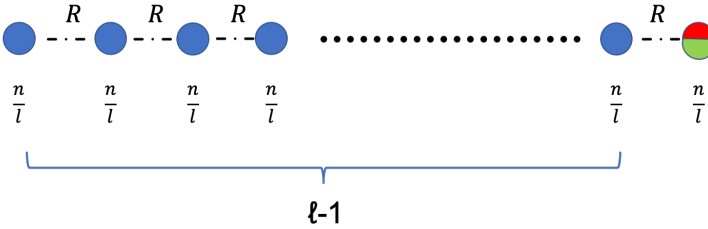

Figure 7: An $\ell$-community instance to show Price of Fairness (**DS**) and incompatibility between **DS** and other fairness constraints.

We restate the following proposition and give its proof:

**Proposition 5.3.** *For any value of $k \geq 2$, imposing **GF** on a solution that only satisfies **DS** can lead to an unbounded increase in the clustering cost even if we allow an additive violation of $\Omega(\frac{n}{k})$.*

*Proof.* The proof follows similarly to Proposition 5.1. For $k \geq 2$ and $k$ is even. Consider the same case as in Figure 5 where we have $\ell = k$. In this case, we set the upper and lower proportion bounds for both **GF** and **DS** to $\frac{1}{2}$. This implies to satisfy **DS** the number of red and blue centers should each be $\frac{k}{2}$. Thus solutions that satisfy the **DS** constraint are the optimal unconstrained solutions as specified in Fact C.3. The rest of the proof proceeds exactly as the proof for the even $k$ case in Proposition 5.1.

For $k \geq 3$ and $k$ is odd, consider the same case as in Figure 6. In this case, we set the upper and lower proportion bounds for **GF** to $\frac{1}{2}$. And we set the upper and lower bound for number of centers in **DS** constraint as $\frac{k-1}{2} + 1$ and $\frac{k-1}{2}$ respectively for both colors. Note that an optimal solution specified in Fact C.3 which chooses either a red or blue point in the right most community as a center satisfy this **DS** constraint. The rest of the proof proceeds exactly as the proof for the odd $k$ case in Proposition 5.1. $\qquad \square$

We restate the following proposition and give its proof:

**Proposition 5.4.** *Imposing **DS** on a solution that only satisfies **GF** leads to a bounded increase in the clustering cost of at most 2 ($\mathrm{PoF} \leq 2$) if we allow an additive violation of 2 in the **GF** constraints.*

*Proof.* This follows from Theorem 4.2 since we can always post-process a solution that only satisfies **GF** into one that satisfies both **GF** at an additive violation of 2 and **DS** simultaneously and clearly from the theorem we would have $\mathrm{PoF} = \frac{\text{clustering cost of \textbf{GF} post-processed solution}}{\text{clustering cost of \textbf{GF} solution}} \leq \frac{2 \text{ clustering cost of \textbf{GF} solution}}{\text{clustering cost of \textbf{GF} solution}} \leq$ 2. $\qquad \square$

# D   Omitted Proofs, Additional Results, and Details for Section 6

In this section, we provide more details and proofs for theorems and facts that appeared in Section 6. We present proof for theorem 6.1. We begin by giving the full definitions of the relevant fairness constraints.

**Definition 2.** *Neighborhood Radius [30]: For a given set of points $\mathcal{C}$ to cluster and a given number of centers $k$, the neighborhood radius of a point $j$ is the minimum radius $r$ such that at least $|\mathcal{C}|/k$ of the points in $\mathcal{C}$ are within distance $r$ of $j$: $NR_{\mathcal{C},k}(j) = \min\{r : |B_r(j) \cap \mathcal{C}| \geq |\mathcal{C}|/k\}$, where $B_r(j)$ is the closed ball of radius $r$ around $j$.*

**Definition 3.** *Fairness in Your Neighborhood Constraint [30]: For a given set of points $\mathcal{C}$ with metric $d(.,.)$, a clustering $(S, \phi)$ is $\alpha_{NR}$-fair if for all $j \in \mathcal{C}$, $d(j, \phi(j)) \leq \alpha_{NR} \cdot NR_{\mathcal{C},k}(j)$.*

**Definition 4.** *Socially Fair [1, 24]: For a clustering problem with $k$ centers on points $\mathcal{C}$ which are from $|\mathcal{H}|$ groups and $\cup_{h \in \mathcal{H}} \mathcal{C}^h = \mathcal{C}$, the socially fair clustering optimization problem is to minimize the maximum average clustering cost across all groups:* $\displaystyle \min_{S:|S| \leq k, \phi} \max_{h \in \mathcal{H}} \frac{1}{|\mathcal{C}^h|} \sum_{j \in C^h} d^p(j, \phi(j))^4.$

Note that socially fair does not optimize over assignment functions because it assumes assignment follows optimal rule: a point is assigned to the cluster center closest to it.

**Definition 5.** *An $\alpha_{SF}$-socially fair solution to a clustering problem is a solution of cost at most $\alpha_{SF}$ of the optimal socially fair solution.*

This definition allows us to bound the clustering cost of an $\alpha_{SF}$-Socially Fair solution $(S_\alpha, \phi_\alpha)$ as:

$$\max_{h \in \mathcal{H}} \frac{1}{|\mathcal{C}^h|} \sum_{j \in C^h} d^p(j, \phi_\alpha(j)) \leq \alpha_{SF} \min_{S:|S| \leq k} \max_{h \in \mathcal{H}} \frac{1}{|\mathcal{C}^h|} \sum_{j \in C^h} d^p(j, \phi(j)).$$

**Definition 6.** *Approximately proportional[15]: Given a set of centers $S \subseteq \mathcal{C}$ with $|S| = k$, it is $\alpha_{AP}$-approximately proportional ($\alpha_{AP}$-proportional) if $\forall U \subseteq \mathcal{C}$ and $|U| \geq \lceil \frac{n}{k} \rceil$ and for all $y \in C$, there exists $i \in U$ with $\alpha_{AP} \cdot d(i, y) \geq d(i, \phi(i))$.*

---

[4]$p = 1$ for the $k$-median and $p = 2$ for the $k$-means

We restate the following theorem and give its proof:

**Theorem 6.1.** *For any value $k \geq 2$, the **fairness in your neighborhood** [30], **socially fair** constraint [1, 24] are each incompatible with **GF** even if we allow an additive violation of $\Omega(\frac{n}{k})$ in the **GF** constraint. For any value $k \geq 5$, the **proportionally fair** constraints [15] is incompatible with **GF** even if we allow an additive violation of $\Omega(\frac{n}{k})$ in the **GF** constraint.*

*Proof.* We show incompatibility of **GF** with the three Fairness notions. Recall that we consider **GF** and another fairness constraint at the same time, and incompatible means there are cases where no feasible solution exists that satisfies both constraints at the same time.

**Lemma 4.** *For any $k \geq 2$, there exist a clustering problem where no feasible solution exists that satisfies both Fairness in Your Neighborhood and **GF** even if we allow an additive violation of $\Omega(\frac{n}{k})$ in the **GF** constraint.*

*Proof.* Consider the case where $k \geq 2$, and consider the clustering problem on a $\ell$-community instance with $k = \ell$. We consider the case where lower and upper proportion bound of **GF** are set to $\frac{1}{2}$ for both colors.

**Claim 5.** *On the above mentioned clustering problem, an $\alpha_{NR}$-fair solution in the Fairness in Your Neighborhood notion for finite $\alpha_{NR}$ is a solution in the set of optimal solutions $(S_{CI}, \phi_{CI})$.*

*Proof.* By Definitions 1 and 2, for any point $j \in \mathcal{C}$, its neighborhood radius is $\mathrm{NR}_{C,k}(j) = \min\{r : |B_r(j) \cap C| \geq |C|/k\} = 0$. This is because each point is in one of the $l$ subset, and by definition, the subset is of size $\frac{n}{l} = \frac{n}{k}$, and points in the same subset are separated by a distance 0.

For a solution on a $\ell$-community instance $(S, \phi) \in (S_{CI}, \phi_{CI})$, for any point $j \in \mathcal{C}$, because $S$ contains a center in the community where $j$ is, $d(j, S) = 0$.

By definition of $\alpha_{NR}$-fairness, this means on a $\ell$-community instance, any solution in $(S_{CI}, \phi_{CI})$ is $\alpha_{NR}$-fair with a finite $\alpha_{NR}$. This is because for any $(S, \phi) \in (S_{CI}, \phi_{CI})$, $d(j, S) = 0 \leq \alpha_{NR} \cdot \mathrm{NR}_{C,k}(j)$ holds for $\alpha_{NR}$ equal to any finite value.

In any solution that is not in $(S_{CI}, \phi_{CI})$, there is at least a point which is assigned to a center not in the point's own community. Thus for such a solution $(S, \phi)$, there exist $j \in \mathcal{C}$, $d(i, S) = R$. Thus for $(S, \phi) \notin (S_{CI}, \phi_{CI})$, for some $j \in \mathcal{C}$, there is no finite $\alpha_{NR}$ such that $d(j, \phi(j)) = R \leq \alpha_{NR} \cdot \mathrm{NR}_{C,k}(j)$ holds.

This shows that a solution that achieves $\alpha_{NR}$-fairness for a finite $\alpha_{NR}$ must be a solution from the set of solutions $(S_{CI}, \phi_{CI})$. $\square$

Thus we have shown that any solution that satisfies the fairness in your neighborhood constraint approximately do not assign points to centers not in its original community.

To characterize the set of solutions that satisfy **GF** with additive $\Omega(\frac{n}{k})$ violation, we consider two cases separately: $k$ is even and $k$ is odd.

Consider the case where $k \geq 2$ is even and refer to Figure 5 where we have $\ell = k$ communities that alternate from red to blue color.

Since the lower and upper proportion bounds are set to $\frac{1}{2}$ for both colors, then to satisfy **GF** each cluster should have both red and blue points. There must exists a cluster $C_i$ of size $|C_i| \geq \frac{n}{k}$, it follows that to satisfy the **GF** constraints at an additive violation of $\rho$, then $|C_i^{\text{blue}}| \geq \frac{1}{2}|C_i| - \rho = \frac{n}{2k} - \rho$ and similarly we would have $|C_i^{\text{red}}| \geq \frac{n}{2k} - \rho$. By setting $\rho = \frac{n}{2k} - \epsilon$ for some constant $\epsilon > 0$, then we have $|C_i^{\text{blue}}|, |C_i^{\text{red}}| > 0$. This implies that a point need be assigned to a center at a distance $R > 0$ for the solution to satisfy **GF** with additive $\Omega(\frac{n}{k})$ violation. Therefore such a solution is not in the solution set that satisfies fairness in your neighborhood.

For a value of $k$ that is odd, see the example Figure 6. Here instead the last community has the same number of red and blue points. We call the cluster whose center is in the last community $C_{\text{last}}$.

If $|C_{\text{last}}| \neq \frac{n}{k}$, then there are points assigned to the center of $C_{\text{last}}$ from other communities incurring cost $R > 0$ or points in the last community are assigned to other centers at distance $R > 0$. In both cases there is at least a point assigned to a center not in its community. If $|C_{\text{last}}| = \frac{n}{k}$, then

in the remaining $k-1$ communities with total of $n - \frac{n}{k}$ points, $k-1$ centers are chosen. A solution satisfying **GF** has one cluster $C_i$ with at least $\frac{1}{k-1}\left(n - \frac{n}{k}\right) = \frac{n}{k}$ points. Then we follow the same argument as in the even $k$ case. That is, to satisfy **GF** with $\rho$ additive violation on the $C_i$, $|C_i^{\text{blue}}| \geq \frac{1}{2}|C_i| - \rho = \frac{n}{2k} - \rho$, $|C_i^{\text{red}}| \geq \frac{n}{2k} - \rho$, with $\rho = \frac{n}{2k} - \epsilon$ for some constant $\epsilon > 0$, at least a point will be assigned to center at distance $R > 0$. Thus such a solution is not in the set of solutions $(S_{CI}, \phi_{CI})$. Thus the set of solutions that satisfies fairness in your neighborhood has no overlap with the set of solutions that satisfies **GF** with $\Omega(\frac{n}{k})$ additive violation. $\square$

**Lemma 5.** *For any $k \geq 2$, there exist a clustering problem where no feasible solution exists that satisfies both Socially Fair and **GF** even if we allow an additive violation of $\Omega(\frac{n}{k})$ to the **GF** constraint.*

*Proof.* We follow a similar line of argument as in Lemma 4. Consider the case when $k \geq 2$, and consider the clustering problem on a $\ell$-community instance with $k = \ell$. We consider the case where lower and upper proportion bound of **GF** are set to $\frac{1}{2}$ for both colors.

**Claim 6.** *On the above mentioned clustering problem, an $\alpha_{SF}$-fair solution in the Socially Fair notion for finite $\alpha_{SF}$ is a solution in the set of optimal solutions $(S_{CI}, \phi_{CI})$.*

*Proof.* Denote the clustering cost of an optimal solution to the a Socially Fair clustering problem as $\text{OPT}_{\text{SF}}$. By definition,

$$\text{OPT}_{\text{SF}} = \min_{S:|S|\leq k} \max_{h\in\mathcal{H}} \frac{1}{|\mathcal{C}^h|} \sum_{j\in C^h} d^p(j, \phi(j)).$$

We can formulate a problem that aims to find an $\alpha_{\text{SF}}$-socially fair solution as a constrained optimization problem. We use a dummy objective function $f$. The constraint can be set up as requiring maximum clustering costs across all colors to be upper-bounded by $\alpha_{\text{SF}}$ times that of the optimal socially fair solution $\text{OPT}_{\text{SF}}$.

The constrained program can be set up as below:

$$\min_{S:|S|\leq k} f$$
$$\text{s.t.} \max_{h\in\mathcal{H}} \frac{1}{|\mathcal{C}^h|} \sum_{j\in C^h} d^p(j, \phi(j)) \leq \alpha_{\text{SF}}\text{OPT}_{\text{SF}}$$

For a clustering problem with $k$ centers on the $\ell$-community instance with $k = \ell$, in a solution $(S, \phi)$ that has one center in each subset, $d(j, \phi(j)) = 0$ for each point $j \in \mathcal{C}$. Thus this solution has clustering cost for each color $h$ as $\sum_{j\in C^h} d^p(j, \phi(j)) = 0$. Which implies that $\text{OPT}_{\text{SF}} = 0$.

Thus on the $\ell$-community instance, feasible solutions to the $\alpha_{\text{SF}}$-socially fair problem, for finite $\alpha_{\text{SF}}$, have $\max_{h\in\mathcal{H}} \frac{1}{|\mathcal{C}^h|}\sum_{j\in C^h} d^p(j, \phi(j)) = 0$. We now show $(S_{CI}, \phi_{CI})$ is the only set of solutions that have $\max_{h\in\mathcal{H}} \sum_{j\in C^h} d^p(j, \phi(j)) = 0$. Thus, they will be the only feasible solutions.

For any solution $(S, \phi)$ that is not in $(S_{CI}, \phi_{CI})$, there must be a point assigned to a center that is not in its own community. For such a point $d(j, \phi(j)) = R$. Thus $\max_{h\in\mathcal{H}} \frac{1}{|\mathcal{C}^h|}\sum_{j\in C^h} d^p(j, \phi(j)) \geq \frac{R}{\max_{h\in\mathcal{H}}|\mathcal{C}^h|}$. Therefore, an $\alpha_{\text{SF}}$-fair solution in the socially fair notion for finite $\alpha_{\text{SF}}$ must be a solution in the set of optimal solutions $(S_{CI}, \phi_{CI})$. $\square$

At this point, similar to proof of Lemma 4, we had shown that any solution that satisfies socially fair approximately do not assign points to centers not in its original community on the $\ell$-community instance. The remaining of the proof is the same as that part of the proof in Lemma 4. We can use the same examples for even $k$ and odd $k$ to show that any solution that satisfies **GF** with $\Omega(\frac{n}{k})$ additive violation any $k \geq 2$ is not in the set of solutions $(S_{CI}, \phi_{CI})$. Thus the set of solutions that satisfies socially fair has no overlap with the set of solutions that satisfies **GF** with $\Omega(\frac{n}{k})$ additive violation. $\square$

**Lemma 6.** *For any $k \geq 5$, there exist a clustering problem where no feasible solution exists that satisfies both Proportional Fairness and **GF** even if we allow an additive violation of $\Omega(\frac{n}{k})$ in the **GF** constraint.*

*Proof.* For a given value of $\alpha_{AP}$ for the proportionally fair constraint, consider Figure 8. For the **GF** constraints, the upper and lower bounds for each color to $\frac{1}{2}$ and the total number of points $n$ is always even. Consider some $k \geq 5$. It follows that the sum of cluster sizes assigned to centers on either the right side or the left side would be at least $\frac{n}{2}$, WLOG assume that it is the left side and denote the total number of points assigned to clusters on the left size by $|C_{LS}|$ and let $S_{LS}$ be the centers on the left side. The total number of points on the left side may not be assigned to a single center but rather distributed among the centers $S_{LS}$. To satisfy the **GF** constraints at an additive violation of $\rho$, it follows that the number of red points that have to be assigned to the left side is at least $\sum_{i \in S_{LS}}(\frac{1}{2}|C_i| - \rho) \geq \frac{n}{4} - k\rho$. Set $\rho = \frac{n}{4k} - \frac{n}{k^2} - 1$, then it follows that at least $\lceil \frac{n}{k} \rceil$ red points are assigned to a center on the left at a distance of at least $R$. Since the maximum distance between any two red points by the triangle inequality is $2r < \frac{R}{\alpha_{AP}}$ it follows that this set of red points forms a blocking coalition. I.e., these points would also have a lower distance from their assigned center if they were instead assigned to a red center. $\qquad\square$

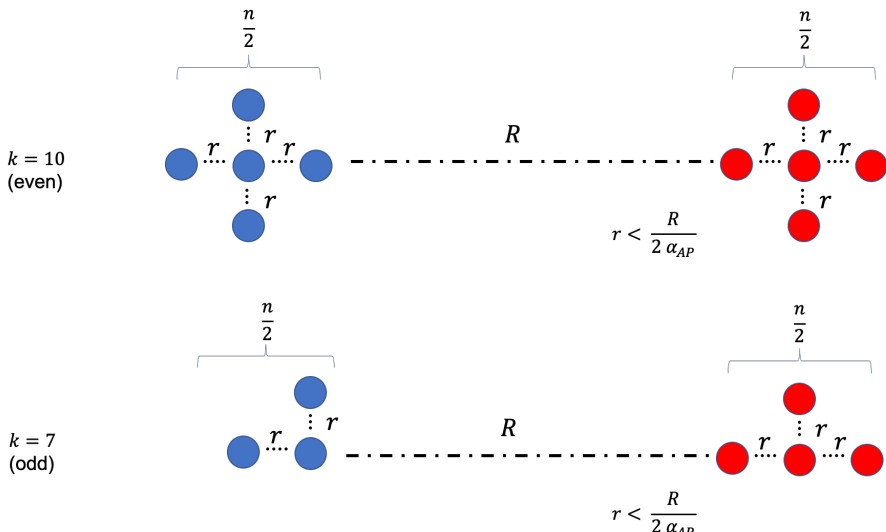

Figure 8: Instances to show incompatibility between Proportional Fairness and **GF**. We always have $n/2$ blue points on the left and $n/2$ red points on the right. For even $k$ we would have $k/2$ locations for the blue and red points each. For odd $k$ we have $\lfloor k/2 \rfloor$ blue locations and $\lceil k/2 \rceil$ red locations. For each color, there is always a location at the center at a distance $r$ from the other locations. Points of different color are at a distance of at least $R$ from each other. For any value of $\alpha_{AP}$ for the proportionally fair constraint, we set $r < \frac{R}{2\alpha_{AP}}$.

$\qquad\square$

We restate the following theorem and give its proof:

**Theorem 6.2.** *For any value $k \geq 3$, the **fairness in your neighborhood** [30], **socially fair** [1, 24] and **proportionally fair** [15] constraints are each incompatible with **DS**.*

*Proof.* Consider a case of the $\ell$-community instance where the first $\ell - 1$ communities consist of points of only blue points. And the last community contains $\frac{n}{2\ell}$ points of red points and $\frac{n}{2\ell}$ points of green color. This $\ell$-community instance is illustrated in Figure 7. Consider the clustering problem where $k = \ell$ and **DS** constraint $k_{\text{blue}}, k_{\text{red}}, k_{\text{green}} > 0$. We establish below two claims.

**Claim 7.** *On the above mentioned clustering problem, an $\alpha_{NR}$-fair solution in the Fairness in Your Neighborhood notion for finite $\alpha_{NR}$ is a solution in the set of optimal solutions $(S_{CI}, \phi_{CI})$.*

**Claim 8.** *On the above mentioned clustering problem, an $\alpha_{SF}$-fair solution in the Socially Fair notion for finite $\alpha_{SF}$ is a solution in the set of optimal solutions $(S_{CI}, \phi_{CI})$.*

Those two claims can be proved with the same argument as in claim 5 and claim 6.

However, satisfying **DS** on this with $k_{\text{blue}}, k_{\text{red}}, k_{\text{green}} > 0$ requires a center of each color be picked. Thus a solution from the set of optimal solutions $(S_{CI}, \phi_{CI})$ does not satisfy **DS** because it will only pick one point from the right most subset as a center. Thus either green points or red points will not appear in the set of centers.

On the other hand, since a solution satisfying **DS** has at least one center of each color, it will contain two centers, one green, one red chosen from the right most subset. And there are $k - 2$ centers allocated to the $k - 1$ communities on the left. By pigeon hole principle, one of the communities of all blue points will have no center allocated. All blue points in this subset are then assigned to a center in a nearby community, thus a **DS** satisfying solution is not in the set $(S_{CI}, \phi_{CI})$. Thus the set of **DS** satisfying solutions has no overlap with the set of solutions that satisfy either one of the two fairness constraints.

Below we use the same example to show incompatibility between **DS** and Proportional Fair.

**Claim 9.** *On the above mentioned clustering problem, there is no feasible solution exists that satisfies both Proportional Fairness and **DS**.*

*Proof.* We show a **DS** satisfying solution on above example is not proportional fair. As argued above, a solution satisfying **DS** can allocate $k - 2$ centers for the $k - 1$ communities on the left. There will be a community of size $\frac{n}{k}$ of which all points are assigned to a nearby center not in its community. This community forms a coalition of size $\frac{n}{k}$ and would have smaller distance if they get assigned a center in their own community. Therefore a **DS** satisfying solution is not proportional fair. □

□

**Remark:** For each of the above proofs we constructed an example which is parametric in the number of centers $k$. Moreover, for these examples for the optimal unconstrained $k$-center objective to equal 0 at least $k$ centers have to be used. I.e., the points are spread over at least $k$ locations. Furthermore, it is not difficult to see in each of the above examples that an optimal solution for the unconstrained $k$-center objective satisfies fairness in your neighborhood, socially fair, and the proportionally fair constraints. In fact, it is easy to show that any $k$-center which has a radius of 0 immediately satisfies fairness in your neighborhood, socially fair, and the proportionally fair constraints. However, the same is not true for **GF** or **DS**. This indicates that the above distance-based fairness constraints can be aligned with the clustering cost whereas the same cannot be said about **GF** or **DS**.

**Compatibility between GF and DS:** One can easily show compatibility between **GF** and **DS**. Specifically, consider some values for the centers over the colors $\{k_h\}_{h \in \mathcal{H}}$ that satisfies the **DS** constraints, i.e. $\forall h \in \mathcal{H} : k_h^l \leq k_h \leq k_h^u$ and has $\sum_{h \in \mathcal{H}} k_h \leq k$. Then simply pick a set $Q_h$ of $k_h$ points of color $h$. Now if we give DIVIDE the entire dataset $\mathcal{C}$ and the set of centers $\cup_{h \in \mathcal{H}} Q_h$ as inputs, i.e. call DIVIDE$(\mathcal{C}, \cup_{h \in \mathcal{H}} Q_h)$, then by the guarantees of divide each center would be active and each cluster would satisfy the **GF** constraints at an additive violation of 2.

Our final conclusions about the incompatibility and compatibility of the constrains are summarized in Figure 9.

# E   Example for Running DIVIDE:

Consider the following example running the DIVIDE subroutine. Specifically, we have a set of points $C$ with a total of $n = |C| = 38$ points. We have 3 colors (blue, red, and green) with the following points: $|C^{\text{blue}}| = 15$, $|C^{\text{red}}| = 14$, and $|C^{\text{green}}| = 9$. We have $Q \subset C$ with a total size of 4 ($|Q| = 4$). Accordingly, we have $T_{\text{blue}} = \frac{15}{4} = 3\frac{3}{4}$, $T_{\text{red}} = \frac{14}{4} = 3\frac{1}{2}$, and $T_{\text{green}} = \frac{9}{4} = 2\frac{1}{4}$. Therefore, in the beginning of the iteration for each color $h$ (line (8) in algorithm block 1) we have $b_{\text{blue}} = 3$, $b_{\text{red}} = 2$, $b_{\text{green}} = 1$. Following the execution of the algorithm, the first three centers $q = 0$ to $q = 2$ receive

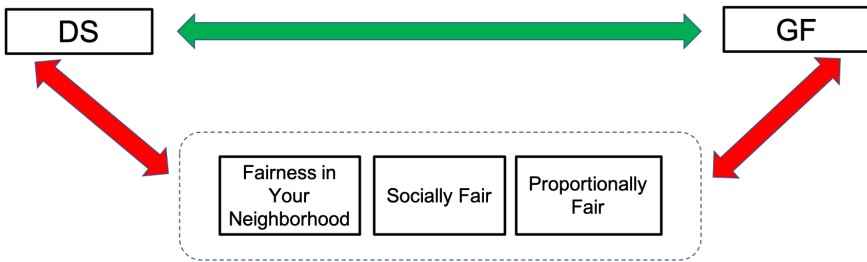

Figure 9: (In)Compatibility of clustering constraints. Red arrows indicate empty feasible set when both constraints are applied, while green arrows indicate non-empty feasibility set when both constraints are applied.

$\lceil T_{\text{blue}} \rceil$ many blue points, the last ($q = |Q| - 1$) and first center ($q = 0$) receive $\lceil T_{\text{red}} \rceil$ many red points, and center $q = 1$ receives $\lceil T_{\text{green}} \rceil$. All other assignments would be the floor of $T_h$. Figure 10 illustrates this.

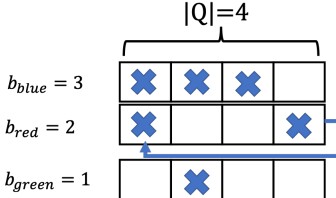

Figure 10: Diagram illustrating how DIVIDE would run over the example. The "tape" has different centers (cells) starting from $q = 0$ and ending with $q = |Q| - 1$. We go over the tape for each color $h \in \mathcal{H}$. In a given row $h$, centers marked with an **X** are assigned $\lceil T_h \rceil$ points, otherwise they are assigned $\lfloor T_h \rfloor$ points.

## F   Additional Experiments Results

Here we show additional experimental results. As a reminder, the lower and upper proportion bounds for any color $h$ to $\alpha_h = (1+\delta)r_h$ and $\beta_h = (1-\delta)r_h$ for some $\delta \in [0,1]$. Further, the **DS** constraints are set to $k_h^l = \lceil \theta r_h k \rceil$ where $\theta \in [0,1]$ and $k_h^u = k$ for every color $h \in \mathcal{H}$.

We call our run over the **Adult** dataset in Section 7 as (**A-Adult**). In that run $\delta = 0.2$ and $\theta = 0.8$. We also, run another experiment (**B-Adult**) over the **Adult** where we set $\delta = 0.05$ and $\theta = 0.9$. Figure 11 shows the new results. We do not see a change qualitatively. It is perhaps noteworthy that the **DS-Violation** values for COLOR-BLIND and ALG-GF are even higher as well as the **GF-Violation** for ALG-DS. On the other hand, we find that our algorithms that satisfy **GF+DS** have very low (almost zero) values for **GF-Violation** and **DS-Violation** at a moderate **PoF** that is comparable to ALG-GF which satisfies only one constraint.

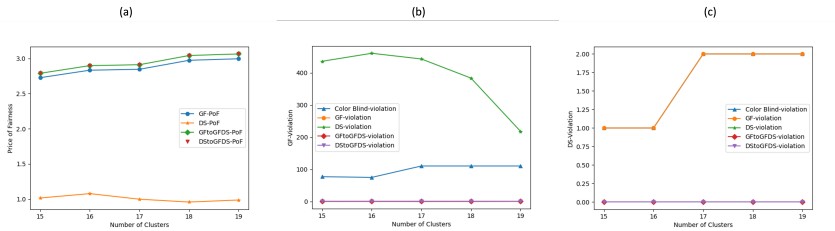

Figure 11: **B-Adult** results: (a) **PoF** comparison of 5 algorithms, with COLOR-BLIND as baseline; (b) **GF-Violation** comparison; (c) **DS-Violation** comparison.

Additionally, we show results over the **Census1990** dataset where we use age as the color (group) membership attribute. As done in [20] we merge the 9 age groups into 3. Specifically, groups $\{0, 1, 2\}$, $\{4, 5, 6\}$, and $\{7, 8\}$ are each merged into one group leading to total of 3 groups. Further, we subsample $6,000$ records from the dataset. We run two experiments where in the first **(A-Census1990)** we have $\delta = 0.05$ and $\theta = 0.7$ whereas in the second **(B-Census1990)** we have $\delta = 0.1$ and $\theta = 0.8$. We also use different cluster values. In terms of the 3 objective measures of **PoF**, **GF-Violation**, and **DS-Violation**, we do not see a qualitative change as can be seen from Figures 12 and 13. Specifically, the **DS** algoruthim (ALG-DS) has a low **PoF** but high **GF-Violation**. Further, COLOR-BLIND and ALG-GF have significant **DS-Violation** values. On the other hand our algorithms for **GF+DS** have low values for both **GF-Violation** and **DS-Violation** and a moderate **PoF**.

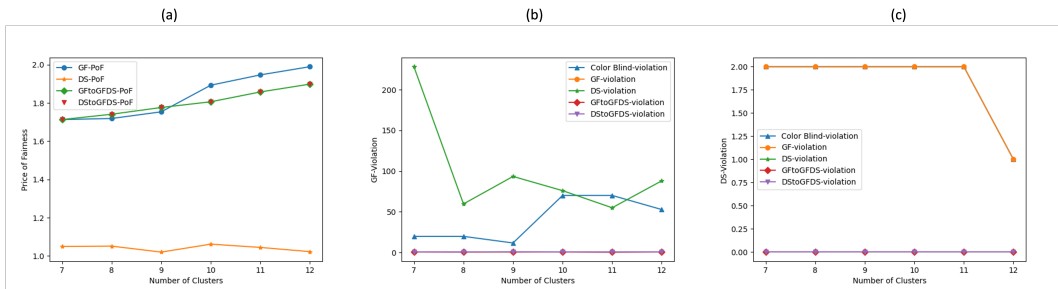

Figure 12: **A-Census1990** results: (a) **PoF** comparison of 5 algorithms, with **COLOR-BLIND** as baseline; (b) **GF-Violation** comparison; (c) **DS-Violation** comparison.

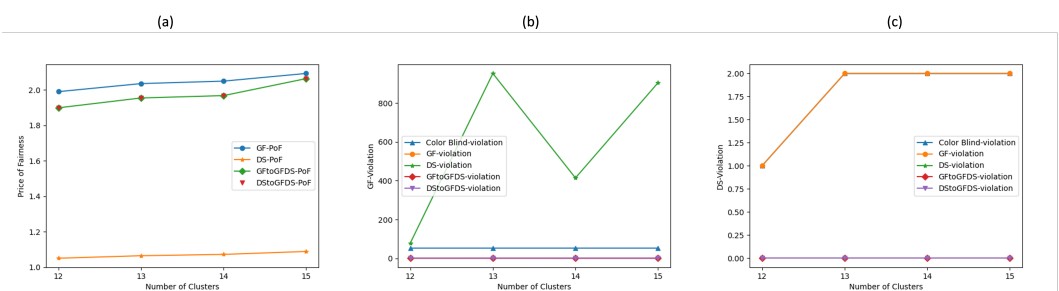

Figure 13: **B-Census1990** results: (a) **PoF** comparison of 5 algorithms, with **COLOR-BLIND** as baseline; (b) **GF-Violation** comparison; (c) **DS-Violation** comparison.

**Run-Time:** Here we show some run-time analysis results. We first calculate the "incremental" run time over **GF**. Specifically, given a solution from the ALG-GF (the algorithm for the **GF** constraints) we see that the additional run-time to post-process it to satisfy the **GF+DS** is very small in proportion. We measure $t_{\textbf{GF}\rightarrow\textbf{GF+DS}} = \frac{\text{Time to process } \textbf{GF} \text{ Solution to satisfy } \textbf{GF+DS}}{\text{Time to obtain } \textbf{GF} \text{ Solution}}$ and show the results in Figure 14 over all 4 runs. We see that the additional run time is constantly at least two orders of magnitude smaller than the time required to obtain a **GF** solution. The fact that added run-time is small further encourages a decision maker to satisfy the **DS** constraint given a solution that satisfies **GF** only.

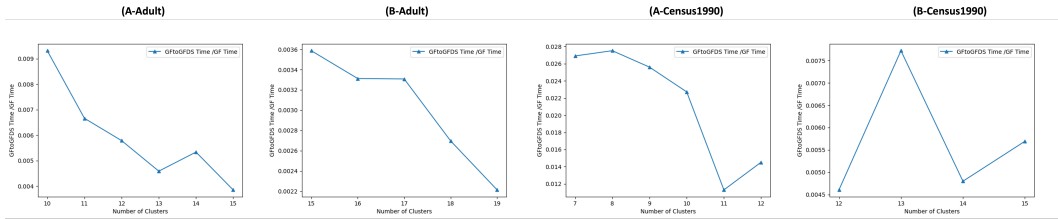

Figure 14: $t_{\textbf{GF}\rightarrow\textbf{GF+DS}}$ over the 4 runs of **(A-Adult)**, **(B-Adult)**, **(A-Census1990)**, and **(B-Census1990)**.

If we were to do the same using the ALG-DS we find the opposite. Specifically, we measure $t_{\mathbf{DS}\rightarrow\mathbf{GF+DS}} = \frac{\text{Time to process } \mathbf{DS} \text{ Solution to satisfy } \mathbf{GF+DS}}{\text{Time to obtain } \mathbf{DS} \text{ Solution}}$ and find that the additional run-time required to satisfy **GF+DS** starting from a **DS** solution is orders of magnitude higher in comparison to the time required to satisfy **DS** as shown in Figure 15. We conjecture that the reason is that ALG-DS is highly optimized in terms of run-time since it runs in $O(nk)$ time [37]. On the other hand, the post-processing step (post-processing a **DS** solution to a **GF+DS**) requires solving an LP which although is done in polynomial time, can be more costly in terms of run time.

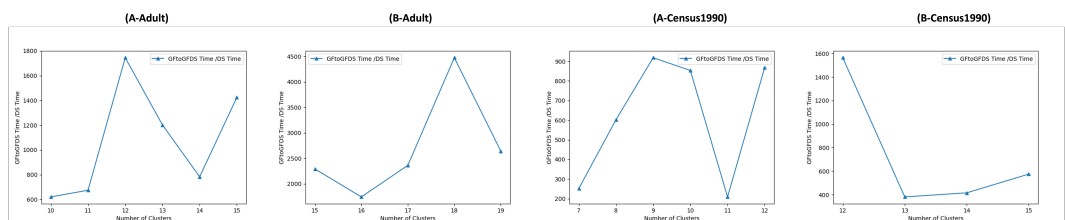

Figure 15: $t_{\mathbf{DS}\rightarrow\mathbf{GF+DS}}$ over the 4 runs of **(A-Adult)**, **(B-Adult)**, **(A-Census1990)**, and **(B-Census1990)**.

Finally, we show a full run-time comparison between GFTOGFDS which starts from a **GF** solution and DSTOGFDS which starts from a **DS** solution. We find that the run-times are generally comparable with one algorithm at times being faster than the other.

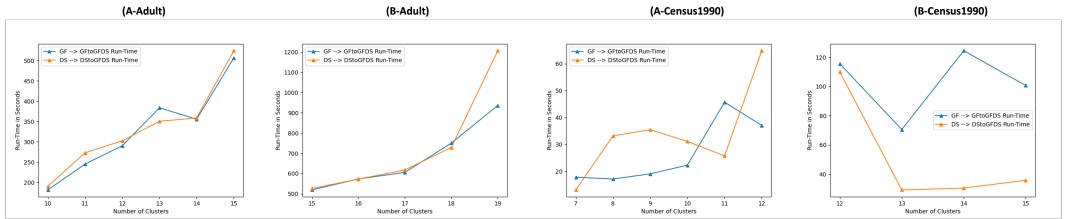

Figure 16: Full run-time comparison between GFTOGFDS and DSTOGFDS over the 4 runs of **(A-Adult)**, **(B-Adult)**, **(A-Census1990)**, and **(B-Census1990)**.

**Using a Bi-Criteria Algorithm as ALG-GF:** Our implementation of **GF** follows Bercea et al. [10] and Bera et al. [9] which would violate the **GF** constraints by at most 2. Empirically, this may cause issues for the GFTOGFDS algorithm since it requires a **GF** algorithm with zero additive violation and therefore assumes that every cluster has at least one point from each color. However, it would not cause issues as long as the resulting solution satisfies condition *of having at least one point from each color in every cluster* which would be the case if $\min_{h\in\mathcal{H}, i\in\bar{S}} \beta_h|\bar{C}_i| > 2$ where $\bar{S}$ is the **GF** solution and $\bar{C}_i$ is its $i^{\text{th}}$ cluster. If the condition not met, then it is reasonable to think that the value of $k$ was set too high or that the dataset includes outlier points since the cluster sizes are very small. Furthermore, using a **GF** algorithm with an additive violation of 2 lead to a final **GF+DS** having a **GF** violation of at most 4 if the condition is satisfied. However, empirically we find the **GF** violation to be generally smaller than 1. Finally, note that we treat the **GF** algorithm as a block-box and therefore it can be replaced by other algorithms such as those of [17, 40] which have no violation for **GF**. In our experiments, we run our algorithms over datasets and value of $k$ where the condition is satisifed.

