# OpenReview forum: "Doubly Constrained Fair Clustering"
_NeurIPS.cc/2023/Conference — NeurIPS 2023 poster_

### Official Review · Reviewer_scu4 · 2023-06-24

**Soundness:** 3 good
**Presentation:** 3 good
**Contribution:** 3 good
**Rating:** 6
**Confidence:** 5

**Summary:**

The paper investigates the fair $k$-center problem and considers the combination of two fairness notions: Group Fairness (GF) and Diversity in Center Selection (DS). The authors show that a constant approximation algorithm for one constraint (GF or DS only) can be extended to a constant approximation algorithm for both constraints simultaneously. Moreover, the authors show that both GF and DS are incompatible with a collection of other distance-based fairness notions.

**Strengths:**

- Due to the importance of fair clustering and multiple fairness notions, it is interesting to study the relationship between different fairness notions.
- The theoretical results show a separation between DP notions (GF and DS), and distance-based notions, which look interesting to me.


**Weaknesses:**

- The studied objective is fair $k$-center, which is less used in machine learning. The extension of the paper to fair $k$-median and fair $k$-means would be more interesting.
- It lacks a discussion of the limitations and future works.

**Questions:**

Could you say anything about fair $k$-median/means?

**Limitations:**

No, the authors should include a discussion on limitations and societal impact.

---

> ### Author Rebuttal · Authors · 2023-08-09
>
> We thank the reviewer for the feedback.
>
> > What about $k$ median and means?
>
> We would like to note that DS for the k-median and k-means was only solved very recently in the paper “Approximation Algorithms for Fair Range Clustering” which appeared in ICML 2023 (a couple of months ago). Therefore, at the time of the writing the problem seemed premature to fully consider.
>
> We note that there have been papers in fair clustering focused on a specific clustering objective such as the k-center (please see e.g. Kleindessner et al 2019, Chiplunkar et al 2020, Jones et al 2020, Harb and Lam 2020) so we do not think that this should be seen as a shortcoming especially given the novelty of the problem which for the first time considers multiple fairness constraints simultaneously in clustering.
>
> Furthermore, we have in fact thought about generalizing the current approach to $k$-median and $k$-means as well. One issue is that not all centers would be active (being assigned a non-empty cluster as described in Section 4.1) and at the same time satisfy GF. However, we believe it is a very interesting problem for future work.
>
> ### References:
>
> M. Kleindessner, P. Awasthi, and J. Morgenstern, ‘‘Fair k-center clus- tering for data summarization,’’ in Proc. Int. Conf. Mach. Learn., 2019, pp. 3448–3457.
>
> A. Chiplunkar, S. Kale, and S. N. Ramamoorthy, ‘‘How to solve fair k- center in massive data models,’’ in Proc. Int. Conf. Mach. Learn., 2020, pp. 1877–1886.
>
> M. Jones, H. Nguyen, and T. Nguyen, ‘‘Fair k-centers via maximum matching,’’ in Proc. Int. Conf. Mach. Learn., 2020, pp. 4940–4949.

---

### Official Review · Reviewer_nxpJ · 2023-06-25

**Soundness:** 3 good
**Presentation:** 3 good
**Contribution:** 2 fair
**Rating:** 6
**Confidence:** 3

**Summary:**

This paper studies how to combine several notions of fairness together in one clustering. Fairness is a popular notion in the context of clustering, however most of previous works had focused on a single notion of fairness at once. This paper studies two specific notions of fairness which are called group fair clustering (GF), and diverse center selection (DF). They give various algorithms that give a constant factor approximation to the k-center problem under the GF+DS constraints (i.e. the clustering has to satisfy both DS and GF constraints). They also study the relationship of these two notions of fairness to other notions that also appear in the literature. More precisely, they show the following.

1)	From a GF fair solution which is a constant factor to the GF problem, one can obtain a constant factor approximation to the GF+DS objective (in polynomial time), assuming a slight violation of the GF constraints.
2)	Similarly, from a DS fair solution which is a constant factor to the DS problem, one can obtain a constant factor approximation to the GF+DS objective (in polynomial time), assuming a slight violation of the GF constraints.
3)	It is also shown that the DS and GF objectives are both incompatible with several other notions of fairness.
4)	They validate their algorithms with experiments.


**Strengths:**

I think this is a conceptually interesting paper. To the best of my knowledge, this is the first paper that combines different fairness criteria together.

**Weaknesses:**

Althought the results are nice, the techniques do not seem very novel.



**Questions:**

Line 2 in pseudocode page 6: the output is stated to satisfy the GF fairness constraint but in fact there is an additive violation. This happens in several other places where the reader first thinks that GF constraint are satisfied exactly without violation. I would encourage to rephrase a little bit.

Typos/minor comments:
Line 138: extra k in equation
Line 211: guarantees


**Limitations:**

yes

---

> ### Author Rebuttal · Authors · 2023-08-09
>
> We thank the reviewer for the feedback. We will fix the typos in lines (138, 211). We will also be explicit about the violation in GF. Furthermore, we believe we presented an elegant solution to the problem and that our modular approach of applying GF or DS algorithms to finally solve both GF+DS can inspire similar future work that considers multiple constraints simultaneously in fair clustering.

---

> > ### Comment · Reviewer_nxpJ · 2023-08-14
> >
> > I thank the authors for their response and clarifications. After reading the rebuttal and other reviews, my assessment has not changed.

---

### Official Review · Reviewer_dBjW · 2023-07-03

**Soundness:** 3 good
**Presentation:** 4 excellent
**Contribution:** 3 good
**Rating:** 6
**Confidence:** 3

**Summary:**

The paper investigates the relationship and intersection between two constraints for fairness clustering: Group Fair, in which populations should be fairly represented in each cluster, and Diversity Selection, in which centers should be fairly selected. It is shown that a solution for one of the constraints can be post-processed into a solution for the other with a constant degradation in results, but also that both constraints are incompatible with other, distance-based fairness constraints introduced in the literature.

**Strengths:**

The theoretical results are very interesting, in terms of showing how it is possible to achieve (with some degradation) both kinds of fairness in clustering but also, and perhaps even more interestingly, when showing how these constraints are incompatible with other fairness settings.

**Weaknesses:**

Experimental results are somewhat weak, with a single dataset being analyzed in the main text and an additional one presented in the supplementary material. The running time of the post-processing method to adapt a DS-compliant solution into a GF+DS-compliant one is very high when compared to the running time of the DS algorithm, but this is only mentioned in the supplementary material as well.

**Questions:**

-  Do you believe the same results would be replicated for other datasets?
- What would happen (in terms of guarantees, results, running times) if the constraints must be guaranteed for more than two groups?
- Were you able to verify the claim in the supplementary material that the overhead in the DS to GF+DS algorithm is due to the necessity of solving an LP?

**Limitations:**

I believe the running time issues should be discussed in the main body of the paper. If there is space for it, it would be interesting to also expand on the consequences of these constraints not being compatible with distance-based ones.

---

> ### Author Rebuttal · Authors · 2023-08-09
>
> We thank the reviewer for the feedback.
>
> > Do you believe the same results would be replicated for other datasets?
>
> We have run the dataset over a reasonable number of datasets as followed in previous fair clustering papers (see e.g, Kleindessner et al 2019, Esmaeili et al 2021). Furthermore, we have added a third dataset (Diabetes), please see the message above and the attached pdf.
>
> > Run-time issue
>
> Before we discuss the run-time it is important to note that the **DS** algorithm we use is from the recent publication of Nguyen et al, 2022. The original run-time of **DS** was in fact $O(n^2)$ (see Chen et al, 2016 “Matroid and knapsack center problems”) and was improved to $O(nk)$ recently by Nguyen et al, 2022 for the general **DS** setting. So a contributing factor was the fact that the **DS** the algorithm we use is very fast, hence not representing a bottleneck. Had we used Chen et al, the conclusion would’ve been very different.
>
> Second, we would argue that there is not really a run-time issue. Since the constraint involves **GF+DS** we are lower bounded in terms of run-time by the max run-time of **GF** and **DS**. The incremental run-time over the GF algorithm is completely negligible (Figure 14), therefore our final solution is not time-consuming. Further, the final run-times shown in Figure 16 are within what is expected for a solution satisfying GF in the literature. So the time for **DS → DS+GF** is not more than what would be considered reasonable.
>
> Our new results on the Diabetes dataset show similar conclusions. We cannot think of a reason why the results would not hold over other datasets.  LP based methods tend to be more time consuming than other combinatorial ``non-LP’’ approaches. This is not a general rule since a non-LP method could possibly have $O(n^5)$ run-time. But for example our combinatorial approach for **GF→ GF+DS** is indeed quite efficient and can be shown to run in linear time.
>
> It is not trivial to satisfy the **GF** for arbitrary bounds through a purely combinatorial approach since the **GF** constraint is in some sense more complicated than **DS** and has to take into account the colors of all of the points in the dataset, not just the selected centers.
>
> Please also see the last section in the response to reviewer Q947 about the run-time as it may help.
>
> > What would happen (in terms of guarantees, results, running times) if the constraints must be guaranteed for more than two groups?
>
> Our algorithms and theoretical guarantees hold for multiple colors. In Appendix F, the census dataset uses more than two groups. Only in Sections 5 and 6 do we show results on two colors, but these results are impossibility results not guarantees of an algorithm. Could the reviewer please clarify this point since can already accommodate more than two groups?
>
>
> ### References:
>
> M. Kleindessner, P. Awasthi, and J. Morgenstern, ‘‘Fair k-center clustering for data summarization,’’ in Proc. Int. Conf. Mach. Learn., 2019, pp. 3448–3457.
>
> S. A. Esmaeili, B. Brubach, A. Srinivasan, and J. P. Dickerson, ‘‘Fair clustering under a bounded cost,’’ 2021, arXiv:2106.07239.
>
> Chen, D. Z., Li, J., Liang, H., and Wang, H. Matroid and knapsack center problems. Algorithmica, 75:27–52, 2016.
> Nguyen, H. L., Nguyen, T., and Jones, M. Fair range k- center. arXiv preprint arXiv:2207.11337, 2022.

---

### Official Review · Reviewer_Q947 · 2023-07-13

**Soundness:** 3 good
**Presentation:** 3 good
**Contribution:** 3 good
**Rating:** 6
**Confidence:** 5

**Summary:**

This paper considers two common notions of fairness in clustering: (I) Group Fairness (GF) and, (II) Diversity in Data Selection (DS). The authors show how to boost an approximate algorithm that satisfies only GF/DF to an approximate algorithm that satisfies them both (with constant violations and constant times cost enlargement). Experiments also support the main claims of the paper.

GF and DF are two popular notions to capture group fairness, I am happy to see that actually they can mostly be satisfied simultaneously without paying much effort. I think the results are an interesting addition to the study of fair clustering.

**Strengths:**

1. The first results link different notions of group fairness constraint in clustering.

2. Algorithms are easy to follow.

**Weaknesses:**

1. The algorithms heavily depend on existing approximate algorithms for clustering with GF or DF. So the result should mostly be regarded as an enhancement of the existing algorithms.

2. The simultaneous guarantees may violate some group constraints. I am worried that in some cases this feature actually contradicts fairness.

**Questions:**

1. What is your running time of algorithms in Theorem 4.1 and 4.2?

**Limitations:**

The algorithms may violate some groups' fairness constraints in the worst case. Will it be an issue in some cases?

---

> ### Author Rebuttal · Authors · 2023-08-09
>
> We thank the reviewer for making these points.
>
> >  The algorithms heavily depend on existing approximate algorithms for clustering with GF or DF. So the result should mostly be regarded as an enhancement of the existing algorithms.
>
> There are a collection of technical details which distinguish our work. First, the notion of active centers (forming non-empty clusters) mentioned in Section 4.1 is trivial to satisfy in **DS** and not needed for **GF**. But once both **GF** and **DS** are considered simultaneously it has to be satisfied but is also not trivial to satisfy since the **GF** constraint implies that points of different colors have to be assigned in the correct proportions to the center. Furthermore, our **Divide** subroutine (Section 4.2) is carefully designed to run in linear time (this is not difficult to see but we will add this and emphasize it in the paper).
>
> Furthermore, since our setting generalizes previously introduced problems, it is perhaps not surprising that it borrows elements from the **GF** and **DS** literature. We consider that to be an advantage since our final algorithm is built in a modular manner which is a much preferable algorithmic design approach. Further, the method of using existing algorithms as subroutines to handle a more complicated variant of the problem appears frequently in the literature. In fact, some **GF** and **DS** algorithms involve a step where an unconstrained (agnostic) clustering algorithm is used as a subroutine and where the output is then post-processed to satisfy **GF** or **DS** (see e.g, Kleindessner et al 2019, Bera et al 2019, Ahmadian 2019, Esmaeili et al 2020).
>
>
> > The simultaneous guarantees may violate some group constraints. I am worried that in some cases this feature actually contradicts fairness.
>
> > The algorithms may violate some groups' fairness constraints in the worst case. Will it be an issue in some cases?
>
> We note that in many fairness papers the fairness constraint might be slightly violated. Note for example some common notions like envy-free-up-to-1-element (EF1) or near-feasible stable matching with couples (Nguyen and Vohra, 2018) where the violation is at most 4 couples, and stability is the fairness notion.
>
>  In our setting we only have a violation for GF but it is always bounded by at most 3 in the worst case. There have been papers for the GF constraint with a similar guarantee. In general, the guarantee is not consequential since a violation of only 3 points is negligible if the size of the cluster is large (consider a cluster of 1,000 points). In fact, if this small violation of 3 points causes an issue, then it would imply that there is a cluster of a very small size. In most practical settings the existence of a very small cluster likely implies that $k$ (number of clusters) was chosen to be too high. Finally, we are not the first to consider such a bounded violation in **GF**, see for example  Bercea et al 2018. Finally, we note that the recent work of Hotegni et al in ICML 2023 has considered violations in **DS** for the $k$-median and means whereas we satisfy them without any violation.
>
> > What is your running time of algorithms in Theorem 4.1 and 4.2?
>
> For Theorem 4.1, note that it uses an LP based algorithm. In general, when LP based methods are used, papers tend not to mention the exact run-time or simply note that it is polynomial. An issue lies in the fact that the optimal run-time for solving LPs remains a topic of ongoing research. Using a known result by Vaidya et al , 1989 would lead to a run-time of $O((nk)^{2.5})$ using his algorithm. However, we use the Simplex algorithm as done in most settings. While the worst case run-time of the Simplex algorithm is exponential, empirically it is very fast. We note that the impressive run-time of the Simplex algorithm had generated interesting work in beyond the worst case analysis. Please see the work of Spielman and Teng on smoothed analysis. One can get an idea about the run-time of the LP from its size (number of variables and constraints).
>
> We will now give a more detailed description about Thorem 4.1’s run-time. In **AssignmentGF** the LP has a total of $nk$ variables with the number of constraints being at most $O(nk)$. The size of the LP is close to or better than some LPs in clustering that can be as large as $O(n^2)$. Note that we use binary search over the values in distance matrix, so we run the LP algorithm $O(log_{}{n})$ which is again standard for the $k$-center problem. Further, the  **MaxFlowGF** algorithm is a special construction of the max flow problem with $|V|=O(n)$ and $|E|=O(nk)$. Using a standard algorithm such as the Ford-Fulkerson or Edmonds-Karp would lead to a run-time of $O(n^2 k)$. We note that this is not slower than previous methods in fair and constrained clustering. The rest of the steps in algorithm 2 are bounded by $O(n)$. Please Appendix F for empirical run-times.
>
> Theorem 4.2 using algorithm 4 takes $O(n)$ a major ingredient in this fast run-time is our efficient **Divide** subroutine. We will add notes about this in the paper.

---

> > ### Comment · Reviewer_Q947 · 2023-08-15
> >
> > Thanks a lot for your detailed response!

---

### Author Rebuttal · Authors · 2023-08-09

We thank the reviewers for their careful reading and constructive criticism.

## Experiments and Datasets:
We note that we have tested our algorithms on two datasets from the UCI repository: Adult shown in Section 7 and Census1990 shown in Appendix F. Further, please see the message below with the attached PDF for new results on the UCI Diabetes dataset. We find that the results for Diabetes are similar qualitatively with the  Adult and Census1990.


## Ethical and Limitation Concerns
The reviewers have raised important points about the limitations of our work and its societal impact. Following the suggestion of reviewer SpBP, we propose the following draft below to be added to the paper to discuss ethical issues and possible limitations:

``Similar to the vast literature in fair clustering, our work is mainly theoretical. We note that Group Fairness and Diversity in Center Selection are two of the most common fairness formalizations in clustering that operationalize legal doctrine such as the notion of disparate impact. There are other fairness formalizations proposed by the fair machine learning community. However, there is still possibly a gap between fairness notions proposed by the research community and industry practitioners in a particular application (see, e.g. Holstein et al. 2019). And in some cases, the stakeholders and practitioners may not fully discern between fair approaches. The application of our algorithms and their effects in a given application would (as most settings in fairness) be a complicated enterprise which should involve significant investigation into the model and the implications of the algorithms as well as consultation with stakeholders.

It is possible that a naive application of a fair algorithm that does not take sufficient considerations into account would result in possible harm. This is not unique to our setting or even fair clustering in general (see e.g., Liu et al, ICML 2018). As an example for possible harm caused by some of the fairness notions we considered, consider the application of **GF** clustering in a biomedical setting. It is possible that agnostic (color-blind) clustering would show correlations between some clusters and a group membership like race. I.e., some groups might be over and/or under represented in some clusters. Through further analysis this cluster might be discovered to be associated with a bad outcome and therefore some groups might be more susceptible than others to this bad outcome. A decision maker would consider this an important discovery since a certain group is overrepresented in the bad outcome cluster. However, the application of fair clustering would make all groups represented in each cluster in close to population level proportions. Therefore, the application of **GF** clustering might not be suited for such an application. One can devise similar examples for **DS** as well. Furthermore, if the decision maker is certain that only **GF** or **DS** is needed then **GF+DS** would not give an advantage and possibly only degrade the clustering cost“.

## Additional Experiments on the Diabetes Dataset

We show additional experimental results on the Diabetes dataset from the UCI repository. This dataset was used in Backurs et al ICML 2019 and Chierichetti et al NeurIPS 2017, we take a sample of 20,0000. Note that only use 1,000 points were used in both papers. Similar to what was done in Backurs et al we use ''age'' and  ''time in hospital'' as the numeric entries where the group membership is ''gender'' which has two values in this dataset. Further, we follow a similar setting to that in the experiments section (Section 7 and F). Specifically, for **GF** the lower and upper proportion bounds for any color $h$ is set to $\alpha_h = (1+\delta) r_h$ and $\beta_h = (1-\delta) r_h$ with $\delta =0.05$. Further, for the **DS** constraints we set $k^l_h = \lceil{\theta r_h k}$ where $\theta \in [0,1]$ and $k^u_h=k$ for every color $h$ with $\theta=0.9$. The results are shown in the figures in the PDF.

Figure 1 shows the **PoF**, **GF-Violation**,and **DS-Violation** similar to previous experiments. We do not see a qualitative change

Figure 2 shows the incremental time. Similar to what we found before the incremental run-time for **GF** is small while it is not for **DS**.

Finally Figure 3 shows the full runtime for both **GF+DS** algorithms and find like on previous datasets that they are comparable.





### References:

Kenneth Holstein, Jennifer Wortman Vaughan, Hal Daum ́e III, Miro Dudik, and Hanna Wallach. Improving fairness in machine learning systems: What do industry practitioners need? In Proceedings of the Conference on Human Factors in Computing Systems (CHI), 2019.

Lydia T. Liu, Sarah Dean, Esther Rolf, Max Simchowitz, and Moritz Hardt. Delayed impact of fair ma- chine learning. In Jennifer Dy and Andreas Krause, editors, Proceedings of the 35th International Conference on Machine Learning, volume 80 of Proceedings of Machine Learning Research, pages 3150–3158, Stock- holmsma ̈ssan, Stockholm Sweden, 10–15 Jul 2018. PMLR.

Arturs Backurs, Piotr Indyk, Krzysztof Onak, Baruch Schieber, Ali Vakilian, and Tal Wagner. Scalable fair clustering. In International Conference on Machine Learning, pages 405–413, 2019.

Flavio Chierichetti, Ravi Kumar, Silvio Lattanzi, and Sergei Vassilvitskii. Fair clustering through fairlets. In Advances in Neural Information Processing Systems, pages 5029–5037, 2017.

---

### Decision · Program_Chairs · 2023-09-21

**Decision:**

Accept (poster)

**Comment:**

This paper considers how to incorporate several notions of fairness into one clustering. The results of the paper were viewed as conceptually interesting and it will be of interest to the community to see how different notions of fairness can be traded off. The technical novelty of the paper was a potential shortcoming, but the results were still interesting.